# Rescuable sleep and synaptogenesis phenotypes in a *Drosophila* model of O-GlcNAc transferase intellectual disability

**Ignacy Czajewski[1], Bijayalaxmi Swain[2], Jiawei Xu[2], Laurin McDowall[1], Andrew T Ferenbach[1,2], Daan MF van Aalten[1,2]***

[1]Division of Molecular, Cell and Developmental Biology, School of Life Sciences, University of Dundee, Dundee, United Kingdom; [2]Section of Neurobiology and DANDRITE, Department of Molecular Biology and Genetics, Aarhus University, Aarhus, Denmark

**Abstract** O-GlcNAcylation is an essential intracellular protein modification mediated by O-GlcNAc transferase (OGT) and O-GlcNAcase (OGA). Recently, missense mutations in *OGT* have been linked to intellectual disability, indicating that this modification is important for the development and functioning of the nervous system. However, the processes that are most sensitive to perturbations in O-GlcNAcylation remain to be identified. Here, we uncover quantifiable phenotypes in the fruit fly *Drosophila melanogaster* carrying a patient-derived OGT mutation in the catalytic domain. Hypo-O-GlcNAcylation leads to defects in synaptogenesis and reduced sleep stability. Both these phenotypes can be partially rescued by genetically or chemically targeting OGA, suggesting that a balance of OGT/OGA activity is required for normal neuronal development and function.

## Editor's evaluation

This important study describes a model for O-GlcNac transferase (OGT) associated Intellectual Disability in *Drosophila*. The authors present convincing data showing that OGT mutant *Drosophila* exhibit defects in neuronal arborisation in development and behaviour (sleep) in adults. These results will be of interest to researchers and clinicians working on protein modifications and intellectual disability.

*For correspondence:
daan@mbg.au.dk

**Competing interest:** The authors declare that no competing interests exist.

## Introduction

Intellectual disability (ID) is a disorder affecting around 1% of the population globally (*McKenzie et al., 2016*), characterised by an intelligence quotient lower than 70 accompanied by reduced adaptive behaviour (*Salvador-Carulla et al., 2011*). Recently, mutations in the X chromosome gene *OGT* were identified as causal for ID, a condition termed OGT congenital disorder of glycosylation (OGT-CDG) (*Willems et al., 2017*; *Pravata et al., 2020b*; *Pravata et al., 2019*; *Pravata et al., 2020a*; *Vaidyanathan et al., 2017*). Patients with OGT-CDG present with diverse signs of varying penetrance, such as microcephaly and white matter abnormalities, as well as non-neurological signs such as clinodactyly, facial dysmorphism, and developmental delay, manifesting as low birth weight and short stature (*Pravata et al., 2020b*). Beyond ID, non-morphological signs of pathogenic *OGT* mutations include behavioural problems as well as sleep abnormalities and epilepsy (*Pravata et al., 2020b*; *Selvan et al., 2018*).

*OGT* encodes a nucleocytoplasmic glycosyltransferase, O-linked β-N-acetyl glucosamine (O-GlcNAc) transferase (OGT), a multifunctional protein composed of two domains: a tetratricopeptide repeat

(TPR) domain and a catalytic domain (**Kreppel and Hart, 1999**; **Lubas et al., 1997**; **Kreppel et al., 1997**). Mutations affecting either domain have been identified in patients with OGT-CDG, though clinical manifestation of the disorder does not appear to segregate with the domain affected (**Pravata et al., 2020b**), suggesting a common disease mechanism. The N-terminal TPR domain is believed to confer substrate specificity for the glycosyltransferase function of OGT (**Iyer and Hart, 2003**; **Levine et al., 2018**; **Clarke et al., 2008**) and is important for non-catalytic functions of the protein (**Urso et al., 2020**; **Levine et al., 2021**). The catalytic domain fulfils two known functions, the transfer of O-GlcNAc onto serine and threonine residues of nucleocytoplasmic proteins (O-GlcNAcylation) **Torres and Hart, 1984**; **Holt et al., 1987**, and the proteolytic activation of Host Cell Factor 1 (HCF-1) (**Capotosti et al., 2011**; **Lazarus et al., 2013**), a known ID-associated protein (**Castro and Quintana, 2020**). While the latter function of OGT potentially contributes to the pathogenicity of some *OGT* mutations (**Pravata et al., 2019**), not all patient mutations have been found to affect HCF-1 processing, neither *in vitro* nor when modelled in stem cells (**Vaidyanathan et al., 2017**; **Selvan et al., 2018**; **Omelková et al., 2023**). Overall, the role of altered O-GlcNAcylation in OGT-CDG pathogenicity remains an open question, as many of the other functions fulfilled by this protein have the potential to contribute to ID.

O-GlcNAcylation is a dynamic modification occurring on around 5000 proteins in the human proteome (**Wulff-Fuentes et al., 2021**). The dynamic nature of the modification is conferred by O-GlcNAcase (OGA), which opposes OGT, catalysing the removal of O-GlcNAc (**Heckel et al., 1998**; **Gao et al., 2001**). O-GlcNAcylation has been extensively implicated in neuronal development, functioning, and disease (**Lee et al., 2021**; **Muha et al., 2021**; **Lagerlöf et al., 2017**; **Olivier-Van Stichelen et al., 2017**; **Chen et al., 2021**; **Kim et al., 2017**) and is therefore likely to play a key role in the pathogenicity of OGT-CDG. The first evidence for the requirement for OGT in development was the study of *Drosophila melanogaster OGT*, *super sex combs* (*sxc*), as a Polycomb group (PcG) gene, amorphic mutations of which were found to result in defects in body segment determination (**Ingham, 1984**), a function later ascribed to its glycosyltransferase activity (**Gambetta and Müller, 2014**). The role of O-GlcNAcylation in PcG function is known to be important for normal neuronal development and highly sensitive to perturbations. For example, maternal hyperglycaemia can drive increased O-GlcNAcylation in the embryo altering neuronal maturation and differentiation patterns through altered PcG function (**Parween et al., 2022**). Multiple additional core developmental regulators have been found to require O-GlcNAcylation for appropriate function, with deregulation of the modification affecting stem cell maintenance through core pluripotency factors such as Sox2 (**Kim et al., 2021a**; **Jang et al., 2012**; **Myers et al., 2016**), cell fate determination through STAT3 (**Fan et al., 2020**), and Notch signalling (**Chen et al., 2021**) and neuronal morphogenesis through the protein kinase A signalling cascade (**Francisco et al., 2009**). Additionally, O-GlcNAcylation is known to play an important role in neuronal functioning related to memory formation (**Muha et al., 2019**; **Taylor et al., 2014**). For example, elevating O-GlcNAcylation in sleep-deprived zebrafish or mice can reverse memory defects associated with a lack of sleep (**Lee et al., 2020**; **Kim et al., 2021b**). The extent of the role of OGT in memory formation is not fully understood, although several proteins important for this process are modulated by O-GlcNAcylation, such as CREB (**Rexach et al., 2012**) or CRMP2 (**Muha et al., 2019**). Therefore, a key unanswered question regarding the aetiology of OGT-CDG is the contribution of the developmental roles of OGT relative to its role in the functioning of the adult nervous system.

With the large number of functionally O-GlcNAcylated proteins and thousands more which remain uncharacterised, identifying the most important processes controlled by O-GlcNAcylation remains challenging. Patient mutations in the catalytic domain present a unique opportunity to better understand processes most sensitive to defective O-GlcNAc cycling. Therefore, we set out to model catalytic domain ID mutations in *Drosophila melanogaster* and characterise their phenotypic effect. *Drosophila* OGT (*Dm*OGT) is highly similar to its human ortholog, with 73% amino acid identity and a high degree of structural similarity (**Mariappa et al., 2015**). However, in the fly OGT does not catalyse HCF-1 proteolytic activation, a function fulfilled instead by taspase 1 (**Capotosti et al., 2007**), eliminating this function of OGT as a confounding variable in understanding the role of O-GlcNAcylation in ID. Previous work modelling OGT-CDG mutations in *Drosophila* has demonstrated that OGT-CDG catalytic domain mutations can reduce global O-GlcNAcylation in adult tissue (**Pravata et al., 2019**), which is linked with defects in habituation and synaptogenesis (**Fenckova et al., 2022**). Here, we demonstrate that a recently discovered ID-associated catalytic domain mutation in *OGT* (resulting in the amino acid substitution C921Y **Omelková et al., 2023**) can reduce O-GlcNAcylation throughout

development in *Drosophila*, which can be rescued by genetically or pharmacologically abolishing or reducing OGA activity, respectively. We find a strong effect of *sxc* mutations on larval neuromuscular junction (NMJ) development, which can be partially reversed by inhibiting or abolishing OGA catalytic activity. Additionally, we demonstrate that a catalytic domain mutation in *sxc* can negatively impact sleep, reducing sleep bout duration. This phenotype can be rescued by abolishing OGA activity and partially reversed by inhibiting OGA in adulthood, suggesting that some aspects of OGT-CDG may not be developmental in origin.

## Results

### An OGT-CDG mutation reduces global O-GlcNAcylation throughout *Drosophila* development

To investigate the contribution of reduced O-GlcNAcylation to phenotypes relevant to OGT-CDG, catalytic domain mutations found in patients were modelled in *Drosophila* using CRISPR-Cas9 mutagenesis. The previously published *sxc*[N595K] (equivalent to human N567K) *Pravata et al., 2019* and the newly generated *sxc*[C941Y] (equivalent to human C921Y) mutant strains were used to assay the effects of OGT-CDG mutations on global O-GlcNAcylation in adult flies. Consistent with previous reports, O-GlcNAcylation in lysates from adult heads was found to be significantly reduced in the *sxc*[N595K] mutant compared to a control genotype (*Figure 1A*; *Pravata et al., 2019*). The newly generated *sxc*[C941Y] mutant strain presented with a significantly more severe reduction in global O-GlcNAcylation, to roughly 40% of the control genotype. This reduction in O-GlcNAcylation was observed despite a modest, yet significant, increase in OGT protein relative to the control genotype. As the reduction in O-GlcNAcylation was modest in the *sxc*[N595K] line, a previously generated catalytically dead mutant strain (*sxc*[K872M]) was further characterised alongside the newly generated *sxc*[C941Y] variant (*Mariappa et al., 2018*), to control for allele-specific effects. The *sxc*[K872M] genotype was previously found to be recessive lethal at the late pupal stages (*Mariappa et al., 2018*); therefore, for this genotype O-GlcNAcylation and OGT levels were only assayed at embryonic and larval stages. Both *sxc*[C941Y] and *sxc*[K872M] stage 16–17 embryos present with significantly reduced O-GlcNAcylation and increased OGT (*Figure 1B*). As *sxc*[K872M] embryos were derived from heterozygous parents, O-GlcNAcylation seen in these embryos is likely largely due to maternally contributed wildtype *sxc* gene product (*Ingham, 1984*; *Sinclair et al., 2009*). By the third-instar larval stage of development, the difference in O-GlcNAcylation between the *sxc*[C941Y] and *sxc*[K872M] genotypes is more pronounced. *sxc*[K872M] larvae present with significantly lower O-GlcNAcylation than both the control and *sxc*[C941Y] genotype (*Figure 1C*). O-GlcNAcylation in the *sxc*[C941Y] larvae remains significantly reduced relative to the control genotype, as at all other stages of development assayed. Surprisingly, at this stage of development, *sxc*[C941Y] larvae do not present with significantly elevated *Dm*OGT protein levels. Strikingly, the mean *Dm*OGT protein levels in *sxc*[K872M] larvae are over eight times higher than in the control genotype.

To determine whether the phenotypic consequences of loss of O-GlcNAc transferase function in the *sxc*[C941Y] mutant flies results in similar phenotypic consequences as a previously characterised *Drosophila* line carrying a hypomorphic mutation in sxc (*sxc*[H537A]), flies were assayed for scutellar bristle development (*Mariappa et al., 2018*). *sxc*[C941Y] flies were found to also present with an increased penetrance of ectopic bristles on the scutellum, with 31% of *sxc*[C941Y] flies presenting with one or more additional bristles, while in the control genotype this only occurred in 8% of flies (*Figure 1—figure supplement 1A*). Taken together, these results demonstrate that hypo-GlcNAcylation due to OGT-CDG variants can be modelled in *Drosophila*. Further supporting the hypothesis that reduced OGT catalytic activity is causal in phenotypes seen in ID, a patient mutation modelled in *Drosophila* results in a similar phenotype as rational mutagenesis of a key *Dm*OGT catalytic residue.

### Pharmacological rescue of O-GlcNAc levels in *sxc*[C941Y] flies

To evaluate whether reduced O-GlcNAcylation in *sxc* mutants with impaired catalytic activity can be rescued to control levels, we sought to elevate O-GlcNAcylation through both genetic and pharmacological means. First, to demonstrate that O-GlcNAcylation can be rescued in flies with impaired *Dm*OGT catalytic activity by abolishing OGA activity, *sxc* mutant flies were crossed with an *Oga* knockout strain (*Oga*[KO]) (*Muha et al., 2020*). When assayed by Western blot, we found that knocking out OGA led to a marked increase in O-GlcNAcylation in lysates from adult heads in the *sxc*[C941Y] line,

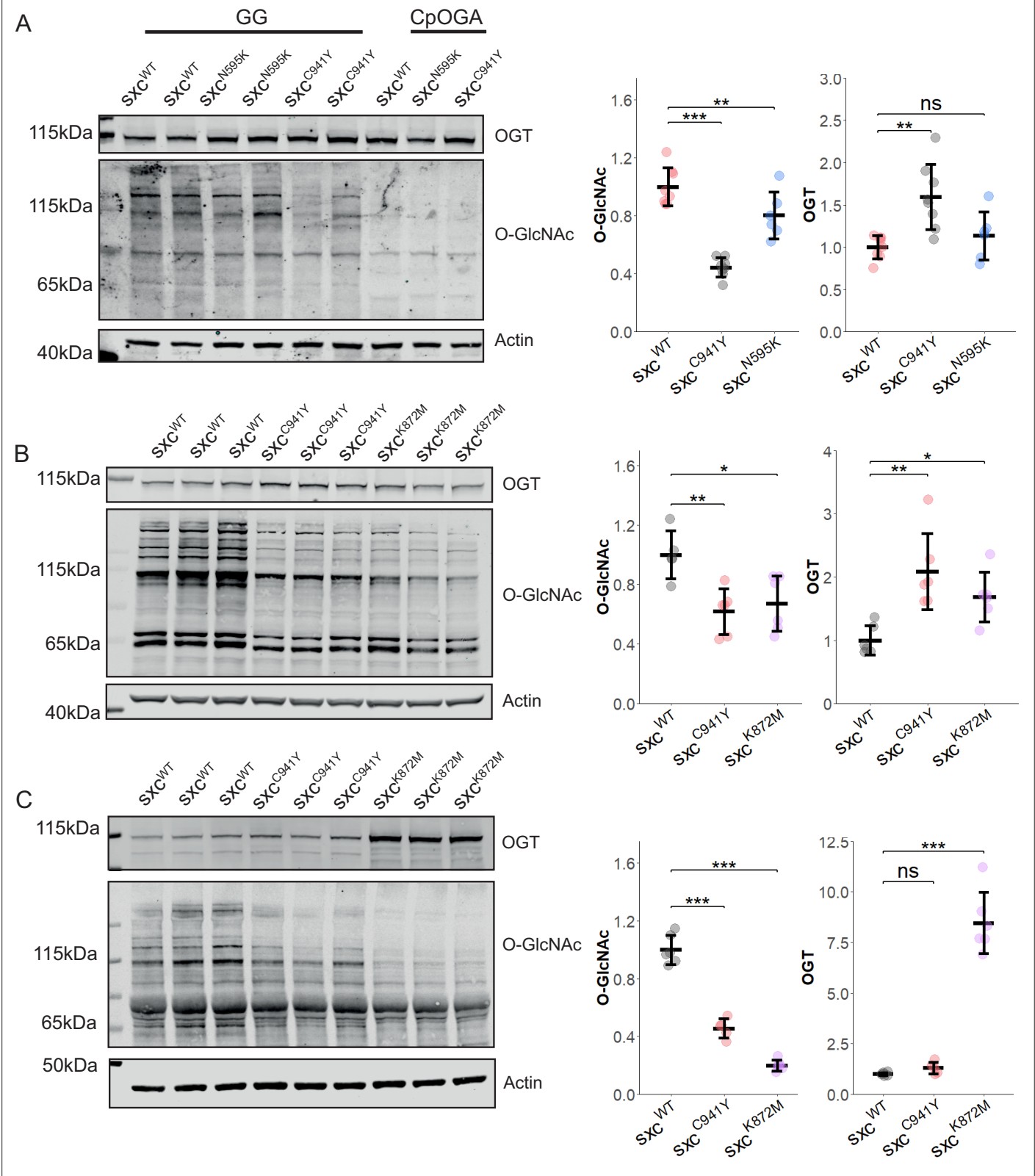

**Figure 1.** Variants affecting the calytic domain of OGT reduce O-GlcNAcylation throughout development. (**A**) Representative western blot of *sxc*[WT] (n = 8), *sxc*[C941Y] (n = 8), and *sxc*[N595K] (n = 6) adult head lysates and quantification (mean ± standard deviation) of OGT and O-GlcNAcylation immunoreactivity normalised to the both the loading and genotype control. *Clostridium perfringens OGA* (CpOGA)-treated lanes demonstrate the specificity of the O-GlcNAc antibody (RL2) used compared to lysates treated with the OGA inhibitor GlcNAc statin G (GG). A significant intergroup difference in O-

*Figure 1 continued on next page*

*Figure 1 continued*

GlcNAcylation was observed (F(2,19) = 42.82, p<0.001), with post hoc analysis revealing a significant reduction in O-GlcNAcylation in both $sxc^{N595K}$ ($p_{adj}$<0.05) and $sxc^{C941Y}$ ($p_{adj}$<0.001) flies relative to the control genotype, and a significant difference between the mutant strains ($p_{adj}$<0.001). A significant intergroup difference was also observed for OGT levels (F(2,19) = 9.137, p<0.01); however, post hoc analysis revealed this was only due to a significant increase in OGT in $sxc^{C941Y}$ flies ($p_{adj}$<0.01). (**B**) Representative western blot of $sxc^{WT}$ (n = 6), $sxc^{C941Y}$ (n = 6), and $sxc^{K872M}$ (n = 6) lysates from stage 16–17 embryos along with OGT and O-GlcNAc quantification. A significant decrease in O-GlcNAcylation (F(2,14) = 8.014, p<0.01) was observed for both $sxc^{C941Y}$ ($p_{adj}$<0.01) and $sxc^{K872M}$ ($p_{adj}$<0.05) embryos, accompanied by a significant increase in OGT (F(2,14) = 9.49, p<0.01) for both genotype ($p_{adj}$<0.01 and $p_{adj}$<0.05, respectively). (**C**) Representative western blot and quantification of lysates from $sxc^{WT}$ (n = 6), $sxc^{C941Y}$ (n = 5), and $sxc^{K872M}$ (n = 6) third-instar larvae, demonstrating a significant decrease in O-GlcNAcylation for both $sxc^{C941Y}$ and $sxc^{K872M}$ larvae (F(2,14) = 184.5, p<0.001, $p_{adj}$<0.001 and $p_{adj}$<0.001, respectively) and a decrease in OGT in $sxc^{K872M}$ larvae (F(2,14) = 122.6, p<0.001, $p_{adj}$<0.001). *p<0.05, **p<0.01, ***p<0.001.

The online version of this article includes the following source data and figure supplement(s) for figure 1:

**Source data 1.** Quantification of OGT and O-GlcNAc immunoreactivity normalised to the loading control and the mean value of the control (*Figure 1A–C*).

**Source data 2.** Raw images of scans of western blots (*Figure 1A–C*).

**Source data 3.** Uncropped scans of western blots with relevant bands/regions indicated (*Figure 1A–C*).

**Figure supplement 1.** Penetrance of supernumerary scutellar bristles in $sxc^{C941Y}$ flies.

**Figure supplement 1—source data 1.** Quantification of scutellar bristle number (*Figure 1—figure supplement 1*).

above levels seen in the control genotype (*Figure 2A*). To assay whether this rescue of O-GlcNAcylation could reverse a phenotype caused by reduced O-GlcNAc transferase activity, we compared the number of scutellar bristles in $sxc^{WT}$, $sxc^{C941Y}$, $sxc^{C941Y}$;$Oga^{KO}$, and $Oga^{KO}$ flies. Surprisingly, we found that despite the $Oga^{KO}$ allele having no effect on its own, $sxc^{C941Y}$;$Oga^{KO}$ flies had an increased penetrance of ectopic scutellar bristles beyond what we observed for $sxc^{C941Y}$ flies (*Figure 1—figure supplement 1B*).

We next set out to identify concentrations at which the OGA inhibitor Thiamet G (TMG) (*Yuzwa et al., 2008*) would restore $sxc^{C941Y}$ global O-GlcNAcylation to control levels. To elevate O-GlcNAcylation in adult *Drosophila*, young adult flies were placed on food supplemented with TMG for 72 hr prior to analysis by western blotting (*Figure 2B*). After assaying varying concentrations of OGA inhibitor, we found that global O-GlcNAcylation was rescued to control levels in $sxc^{C941Y}$ flies fed 3 mM TMG for 72 hr. Paradoxically, a higher 5 mM concentration did not have the same effect. Flies fed this higher concentration of TMG were found to have significantly decreased global O-GlcNAcylation relative to the control genotype, though this appeared to be due to an alteration in the pattern of O-GlcNAcylation with some substrates retaining elevated O-GlcNAcylation relative to $sxc^{C941Y}$ flies fed standard food (*Figure 2—figure supplement 1*). Accompanying elevated O-GlcNAcylation, TMG treatment resulted in decreased levels of *Dm*OGT. For $sxc^{C941Y}$ flies fed 3 mM TMG, *Dm*OGT protein levels were rescued to control levels, while for flies fed 5 mM TMG, *Dm*OGT decreased below levels seen in the control genotype. To assay whether the same pharmacological rescue could be performed during development, adults were allowed to lay eggs on food supplemented with TMG and the O-GlcNAcylation levels of their offspring were measured by western blot at the wandering third-instar stage (*Figure 2C*). Presumably due to differences in feeding behaviour, TMG concentrations required to rescue O-GlcNAcylation during the larval stages of development were much lower than for adults. At 150 µM TMG, O-GlcNAcylation in $sxc^{C941Y}$ larvae was no longer significantly different from the control genotype, while O-GlcNAcylation in larvae fed 200 µM TMG was both significantly higher than in the $sxc^{C941Y}$ larvae fed standard food and not significantly different from the control genotype. Overall, these results demonstrate that defective O-GlcNAc homeostasis in flies carrying an OGT-CDG mutation can be restored by reducing OGA activity through pharmacological inhibition.

## $sxc^{C941Y}$ flies possess a NMJ bouton phenotype

Previous research has identified an important role for O-GlcNAcylation in excitatory synapse function (*Lagerlöf et al., 2017*; *Fenckova et al., 2022*; *Muha et al., 2020*). To ascertain the contribution of this role of O-GlcNAcylation to ID, synaptic development was assayed at the larval NMJ. This synapse is an established model for mammalian central nervous system excitatory synapses and has been previously used to study the role of genes implicated in ID (*Pan et al., 2004*). To assay the effects of *sxc* mutations on NMJ morphology, type 1b NMJs of muscle 4 were visualised by immunostaining for

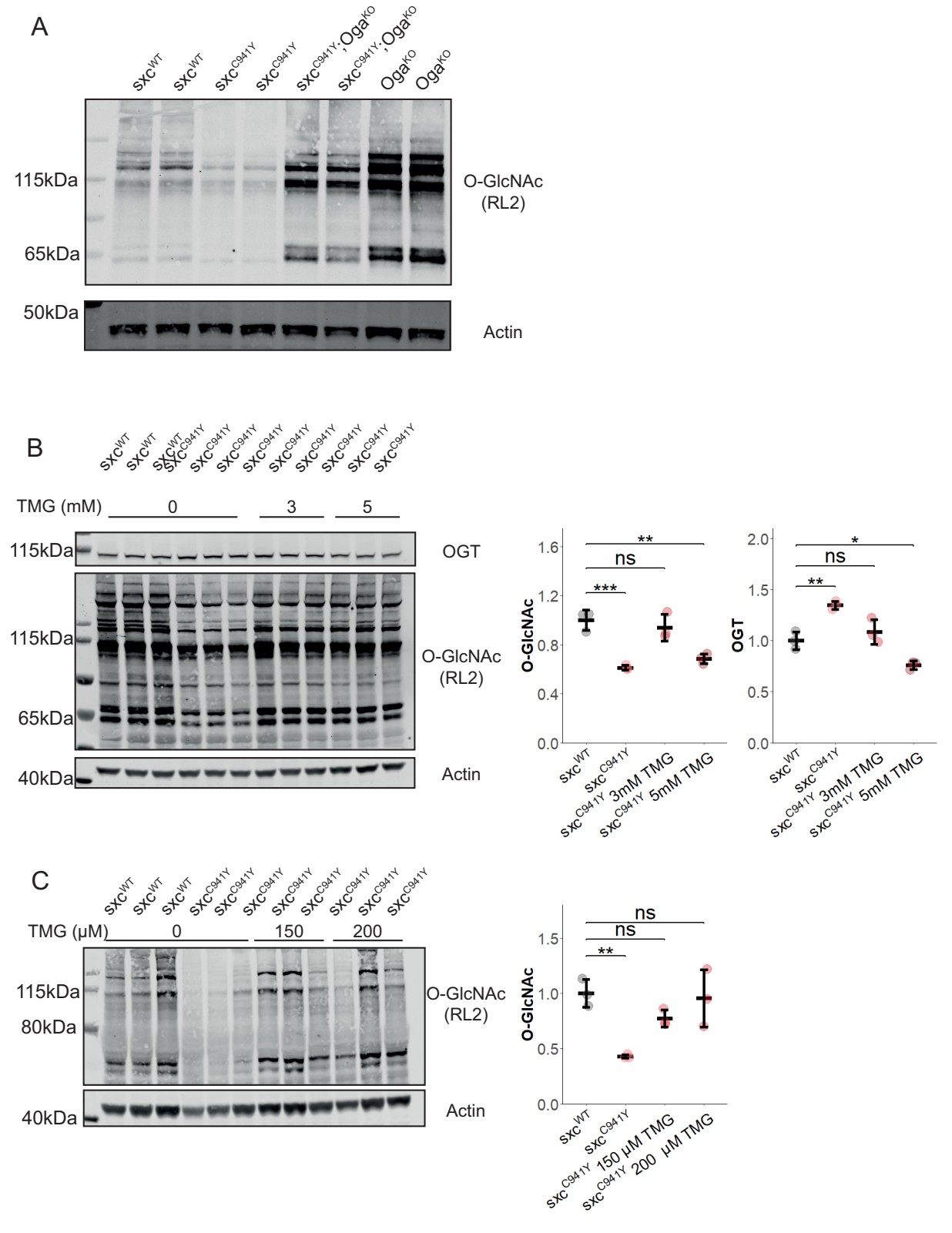

**Figure 2.** Genetic and phamacological rescue of O-GlcNAcylation in *sxc^C941Y^* flies. (**A**) Representative western blot (of three) of *sxc^WT^*, *sxc^C941Y^*, *sxc^C941Y^;Oga^KO^* and *Oga^KO^* adult head lysates, immunolabelled with RL2 to detect O-GlcNAcylation and actin as a loading control. (**B**) Western blot and quantification of adult head lysates of *sxc^WT^*, *sxc^C941Y^* vehicle, *sxc^C941Y^* fed 3 mM Thiamet G (TMG) and *sxc^C941Y^* fed 5 mM TMG (n = 3), immunolabelled for O-GlcNAcylation using the RL2 antibody, OGT, and actin as a loading control. A significant intergroup difference was observed for both O-

*Figure 2 continued on next page*

*Figure 2 continued*

GlcNAcylation ($F_{(3,8)}$ = 20.86, p<0.001) and OGT ($F_{(3,8)}$ = 27.28, p<0.001) levels, with post hoc analysis revealing that O-GlcNAcylation and OGT levels were not significantly different between $sxc^{WT}$ flies and $sxc^{C941Y}$ flies fed 3 mM TMG ($p_{adj}$=0.75 and $p_{adj}$=0.57, respectively). Both $sxc^{C941Y}$ flies fed a vehicle control and 5 mM TMG present with significantly different O-GlcNAcylation ($p_{adj}$<0.001 and $p_{adj}$<0.01, respectively) and OGT ($p_{adj}$<0.01 and $p_{adj}$<0.05, respectively) levels. (**C**) Western blot and quantification of third-instar larval lysates of $sxc^{WT}$, $sxc^{C941Y}$ vehicle, $sxc^{C941Y}$ fed 150 µM TMG and $sxc^{C941Y}$ fed 200 µM TMG (n = 3), immunolabelled for O-GlcNAcylation using the RL2 antibody and actin as a loading control. O-GlcNAcylation significantly differed between groups ($F_{(3,8)}$ = 9.11, p<0.01), with both 150 µM and 200 µM TMG rescuing O-GlcNAcylation levels in $sxc^{C941Y}$ larvae to be no longer significantly different relative to the control genotype ($p_{adj}$=0.31 and $p_{adj}$=0.98, respectively) and 200 µM TMG treatment significantly elevating O-GlcNAcylation relative to the untreated $sxc^{C941Y}$ larvae ($p_{adj}$<0.05). *p<0.05, **p<0.01, ***p<0.001.

The online version of this article includes the following source data and figure supplement(s) for figure 2:

**Source data 1.** Raw images of scans of Western blots (*Figure 2A–C*).

**Source data 2.** Uncropped scans of Western blots with relevant bands/regions indicated (*Figure 2A–C*).

**Figure supplement 1.** Non-isometric increase in O-GlcNAcylation in Thiamet G fed flies.

the subsynaptic reticulum protein Discs large 1 (Dlg1) (*Gan and Zhang, 2018*) and with an anti-HRP antibody to visualise neuronal membranes (*Fabini et al., 2001*; *Figure 3A*). Upon quantification with a semiautomated ImageJ macro (*Nijhof et al., 2016*), several parameters measured were found to significantly differ between the NMJs in control genotype larvae and $sxc^{C941Y}$ and the catalytically dead $sxc^{K872M}$ larvae. The average NMJ area in $sxc^{WT}$ larvae (mean ± standard deviation, 326 ± 52 µm$^2$) was significantly higher than in both $sxc^{C941Y}$ (278 ± 28 µm$^2$) and $sxc^{K872M}$ mutant larvae (198 ± 32 µm$^2$), with a significant difference between the two $sxc$ mutant groups. This phenotype was partially rescued in the $sxc^{C941Y}$;$Oga^{KO}$ line (291 ± 39 µm$^2$), relative to the control genotype, although the total area of the NMJs was not affected in the $Oga^{KO}$ larvae (337 ± 32 µm$^2$), consistent with previous research on $Oga^{KO}$ larvae (*Fenckova et al., 2022*; *Figure 3B*). Total length was also significantly different between the control genotype (mean ± standard deviation, 115 ± 18 µm) and $sxc^{C941Y}$ (93 ± 11 µm) and $sxc^{K872M}$ larvae (74 ± 7 µm). This parameter was also partially rescued in $sxc^{C941Y}$;$Oga^{KO}$ larvae (102 ± 14 µm) relative to $sxc^{WT}$ larvae, while being unaffected in the $Oga^{KO}$ genotype (115 ± 13 µm) (*Figure 3C*). Finally, bouton numbers were also significantly reduced in both $sxc^{C941Y}$ (mean ± standard deviation, 15 ± 3) and $sxc^{K872M}$ (12 ± 1) larvae, relative to the $sxc^{WT}$ controls (19 ± 3). Unlike total area and length, this parameter remained significantly reduced in the $sxc^{C941Y}$;$Oga^{KO}$ line (16 ± 2) relative to the control genotype (*Figure 3D*).

As O-GlcNAcylation has been shown to regulate overall body size (*Sekine et al., 2010*; *Park et al., 2011*) and NMJ area correlates with muscle size (*Nijhof et al., 2016*), we decided to measure muscle size in $sxc^{K872M}$ larvae to determine whether changes in overall body growth could explain the NMJ phenotype we observed. No significant difference in muscle size was observed between $sxc^{WT}$ and $sxc^{K872M}$ larvae, and when NMJ area was normalised to muscle area, this parameter remained significantly reduced in $sxc^{K872M}$ larvae (*Figure 3—figure supplement 1A*). Further, to ascertain whether loss of normal O-GlcNAcylation impairs NMJ bouton growth through pre-synaptic or post-synaptic mechanisms, wild type $sxc$ was overexpressed either in neurons (elavL3-Gal4) or muscles (mhc-Gal4), in a $Dm$OGT catalytically dead background ($sxc^{K872M}$). This demonstrated that overexpression of wild type $sxc$ in $sxc^{K872M}$ larval neurons could significantly increase NMJ total area, but not length or bouton number. By contrast, similar overexpression in muscle cells did not lead to any significant effect on NMJ morphology (*Figure 3—figure supplement 2*). Overall, growth of larval NMJs is broadly stunted in larvae modelling OGT-CDG and in larvae completely lacking OGT catalytic activity, with the phenotype partially rescued in the former by knocking out $Oga$. This is at odds with previously published research, which shows that both rationally designed hypomorphic mutants and ID mutations in the TPR domain result in increased growth at the NMJ (*Fenckova et al., 2022*). To address this disparity, we measured NMJ parameters in larvae of one of the genotypes previously assayed, $sxc^{H596F}$. We found that this mutation also results in a significant decrease in NMJ area (mean ± standard deviation, 260 ± 23 µm$^2$) relative to the control genotype (304 ± 18 µm$^2$), with a similar effect for length and bouton number, consistent with the other genotypes assayed here (*Figure 3—figure supplement 1B–E*).

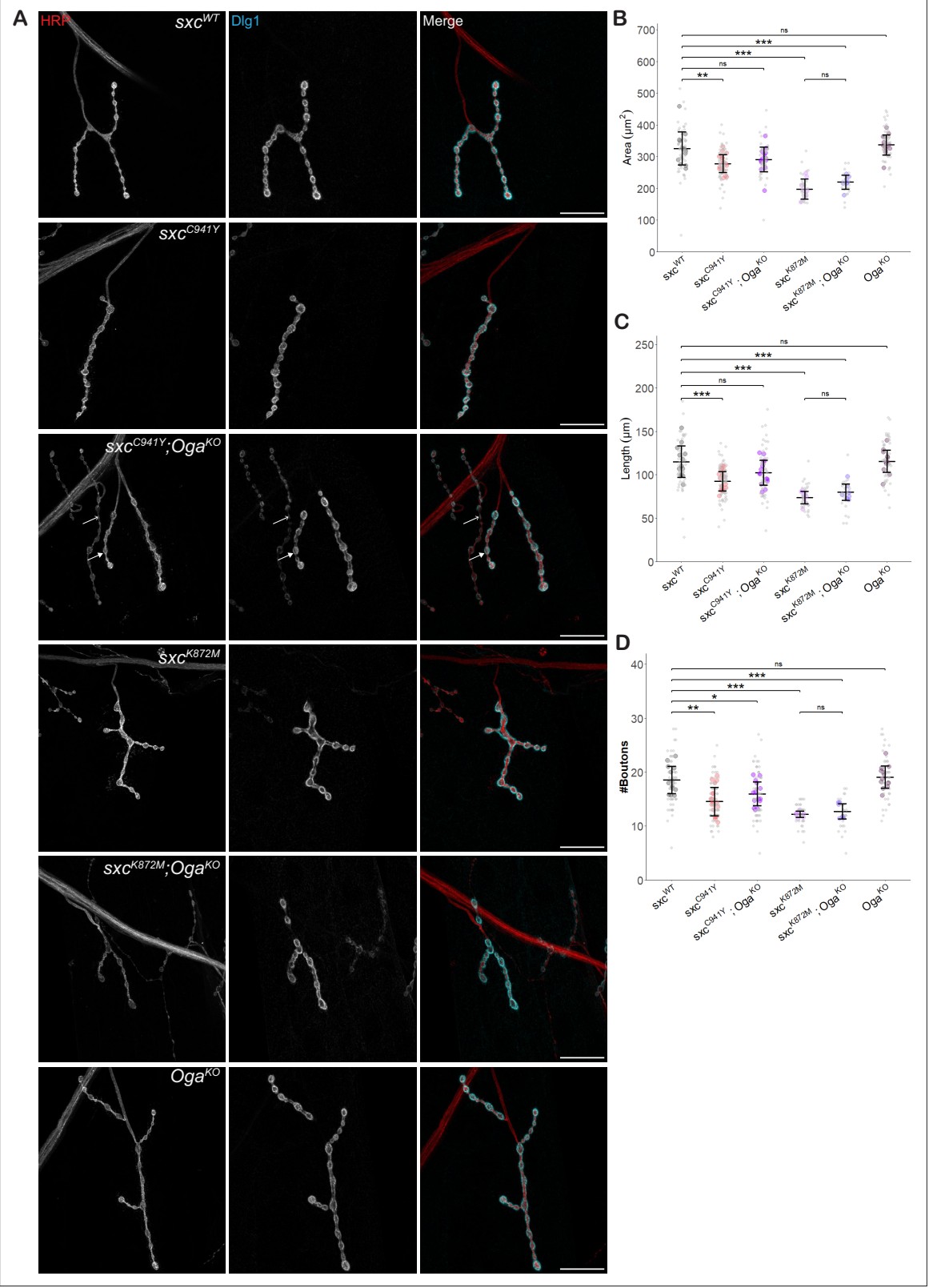

**Figure 3.** Variants affecting the catalytic domain of OGT impair neuromuscular junction development. (**A**) Representative images of larval neuromuscular junctions (NMJs) immunolabelled with anti-HRP (red), anti-Discs Large 1 (cyan) and both (scale bars 25 μm) for *sxc^WT^* (n = 13), *sxc^C941Y^* (n = 19), *sxc^C941Y^;Oga^KO^* (n = 14), *sxc^K872M^* (n = 8), *sxc^K872M^;Oga^KO^* (n = 7), and *Oga^KO^* (n = 13) larvae. In the *sxc^C941Y^;Oga^KO^* panel, the closed arrow indicates 1b boutons, analysed here, while the open arrow indicates an example of 1 s boutons, not analysed here. (**B**) Quantification of NMJ area (mean ± SD),

*Figure 3 continued on next page*

*Figure 3 continued*

which was found to be significantly different between genotypes (F(5,68) = 23.05, p<0.001). Relative to the *sxc*[WT] control, both *sxc*[C941Y] and *sxc*[K872M] larvae presented with a smaller NMJ area ($p_{adj}$<0.01 and $p_{adj}$<0.001, respectively), which was partially rescued in the *sxc*[C941Y]*;Oga*[KO] strain ($p_{adj}$=0.14), though the *Oga*[KO] larvae did not present with a significantly increased NMJ area ($p_{adj}$=0.97). (**C**) Quantification of NMJ length (mean ± SD), which was found to be significantly different between genotypes (F(5,68) = 17.75, p<0.001). Relative to the control genotype, both *sxc*[C941Y] and *sxc*[K872M] larvae presented with overall shorter NMJ length ($p_{adj}$<0.001 for both), while NMJ length was not significantly different in *sxc*[C941Y]*;Oga*[KO] larvae ($p_{adj}$=0.13), despite *Oga*[KO] NMJ length not being affected ($p_{adj}$=0.99). (**D**) Bouton number (mean ± SD) is significantly reduced in *sxc*[C941Y] and *sxc*[K872M] larvae (F(5,68) = 18.11, p<0.001, $p_{adj}$<0.001 for both), and remains significantly reduced in *sxc*[C941Y]*;Oga*[KO] larvae ($p_{adj}$<0.05). Values for individual NMJs are represented as small grey points, with averages for each larva represented as larger coloured points. Descriptive and inferential statistics were performed on larval averages, *p<0.05, **p<0.01, ***p<0.001.

The online version of this article includes the following source data and figure supplement(s) for figure 3:

**Source data 1.** Neuromuscular junction morphological parameters (*Figure 3B–D*).

**Figure supplement 1.** Reduced neuromuscular junction growth is not due to reduced muscle size nor allele dependent.

**Figure supplement 1—source data 1.** Neuromuscular junction morphological parameters (*Figure 3—figure supplement 1A–E*).

**Figure supplement 2.** Neuronal overexpression of *sxc* partially rescues neuromuscular junction defects caused by loss of OGT catalytic activity.

**Figure supplement 2—source data 1.** Neuromuscular junction morphological parameters (*Figure 3—figure supplement 2B–D*).

## Pharmacological rescue of OGT-CDG NMJ phenotypes

To determine whether the (partial) rescue of NMJ parameters by genetic ablation of OGA activity can be recapitulated by pharmacological means, larvae were fed 200 µM TMG to elevate O-GlcNAc-ylation to control levels, as previously determined (*Figure 2C*). As with knocking out *Oga*, elevating O-GlcNAcylation pharmacologically resulted in a partial rescue of NMJ parameters (*Figure 4A*). The total NMJ area in *sxc*[C941Y] larvae treated with 200 µM TMG (mean ± standard deviation, 303 ± 40 µm²) was no longer significantly different relative to the control genotype (319 ± 31 µm²) while *sxc*[C941Y] fed a vehicle control presented with reduced NMJ area relative to the control genotype (272 + 40 µm²) (*Figure 4B*). Unlike in *sxc*[C941Y]*;Oga*[KO] larvae, TMG inhibition in *sxc*[C941Y] larvae did not significantly rescue NMJ length (median ± interquartile range, 106 ± 13 µm) relative to the control genotype (117 ± 9 µm), although a non-significant increase in length relative to *sxc*[C941Y] larvae fed a vehicle was observed (95 µm ± 8) (*Figure 4C*). Similar to the OGA knockout experiment (*Figure 3D*), *sxc*[C941Y] larvae fed 200 µM TMG presented with significantly fewer boutons per NMJ (mean ± standard deviation, 16 ± 2) relative to the control genotype (19 ± 2) without a significant difference relative to the *sxc*[C941Y] larvae fed a vehicle control (15 ± 2) (*Figure 4D*). Overall, this demonstrates that pharmacological inhibition of OGA activity can partially rescue synaptogenesis in OGT-CDG mutant larvae.

## Fragmented sleep in *sxc*[C941Y] flies is reversible by normalising global O-GlcNAcylation

Patients with ID present with hyper-activity and sleep disturbances more often than the general population (*Faraone et al., 2017*; *Köse et al., 2017*). Several patients affected by OGT-CDG follow this pattern, presenting with sleep disturbances and behavioural abnormalities (*Pravata et al., 2020b*; *Selvan et al., 2018*). To assay whether activity and sleep are also disrupted in a *Drosophila* model of OGT-CDG, we used the *Drosophila* Activity Monitor (DAM) to measure these parameters (*Figure 5A and B*). In *Drosophila* research, sleep is commonly defined as a period of five or more minutes of quiescence, which is accurately measured by the DAM system (*Donelson et al., 2012*). Total activity of *sxc*[C941Y] flies (median ± interquartile range, 1.25e3 ± 6.6e2 counts/24 hr) was not significantly different from the control genotype (1.23e3 ± 5.8e2 counts/24 hr). However, *sxc*[C941Y]*;Oga*[KO] flies were significantly less active than the control genotype (8.9e2 ± 3.9e2 counts/24 hr), despite the *Oga*[KO] allele having no effect on total activity on its own (1.13e3 ± 5.7e2 counts/24 hr) (*Figure 5C*). By contrast, *sxc*[C941Y] flies did present with reduced total sleep (mean ± standard deviation, 8.1e2 ± 1.8e2 min/24 hr), relative to the control genotype (9.4e2 ± 1.4e2 min/24 hr), which was rescued in *sxc*[C941Y]*;Oga*[KO] flies to wild type levels (9.7e2 ± 1.3e2 min/24 hr) (*Figure 5D*). Upon more detailed investigation, the nature of sleep disruption in the OGT-CDG flies was found to be due to a reduced duration of individual sleep bouts both during the day and night in these flies (median ± interquartile range, 28 ± 13 min and 39 ± 30 min, respectively) compared to the control genotype (42 ± 22 min and 75 ± 56 min, respectively). Mean sleep bout duration in *sxc*[C941Y] flies is partially rescued by elevating

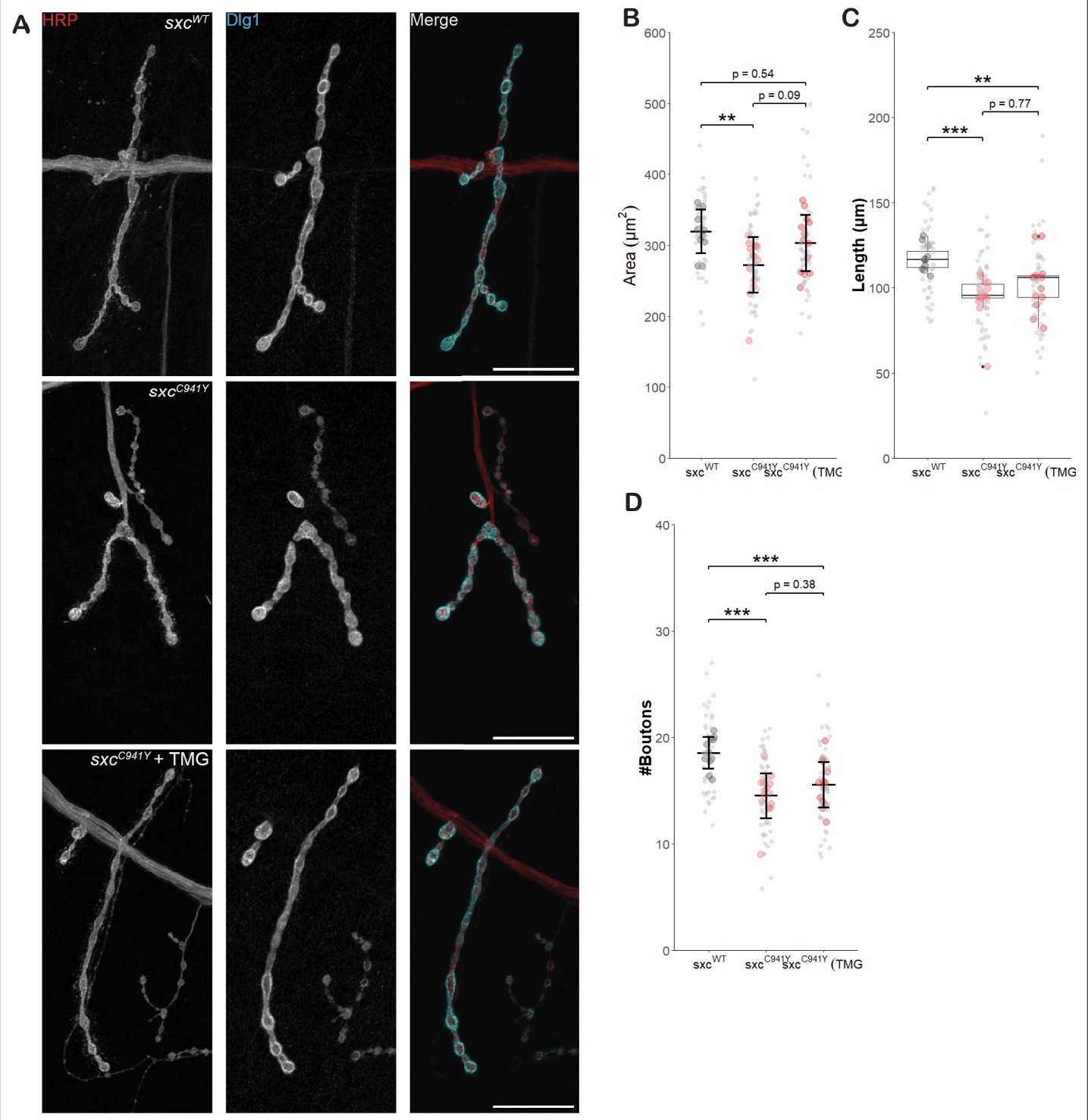

**Figure 4.** Thiamet G partially rescues neuromuscular junction defects in *sxc^C941Y* larvae. (**A**) Representative images of neuromuscular junctions (NMJs) immunolabelled with anti-HRP (red), anti-Discs Large 1 (cyan), and both (scale bars 25 µm) for *sxc^WT* (n = 11), *sxc^C941Y* (n = 14), and *sxc^C941Y* fed 200 µM TMG (n = 13). (**B**) NMJ area (mean ± SD) is significantly reduced in *sxc^C941Y* larvae fed a vehicle control (F(2,35) = 5.264, p=0.01, $p_{adj}$<0.01), relative *sxc^WT* larvae. *sxc^C941Y* larvae fed 200 µM TMG no longer present with a significant reduction in total NMJ area ($p_{adj}$=0.54). (**C**) Total NMJ length (median ± IQR) is significantly different between groups ($X^2$(2) = 17.483, p<0.001); however, unlike total area, post hoc analysis demonstrates that this parameter remains significantly reduced compared to the *sxc^WT* control for both vehicle and TMG treated *sxc^C941Y* larvae ($p_{adj}$<0.001 and $p_{adj}$<0.01, respectively). (**D**) Quantification of bouton number (mean ± SD) demonstrated a significant intergroup difference (F(2,35) = 13.6, p<0.001), with both vehicle and TMG fed *sxc^C941Y* larvae presenting with significantly reduced bouton number ($p_{adj}$<0.001 and $p_{adj}$<0.01, respectively). Values for individual NMJs represented

*Figure 4 continued on next page*

*Figure 4 continued*

as small grey points, with averages for each larva represented as larger coloured points. Descriptive and inferential statistics were performed on larval averages, *p<0.05, **p<0.01, ***p<0.001.

The online version of this article includes the following source data for figure 4:

**Source data 1.** Neuromuscular junction morphological parameters (*Figure 4B–D*).

global O-GlcNAcylation through knocking out *Oga* both during the day (32 ± 19 min) and at night (56 ± 40 min), although during both time periods sleep bout duration remained significantly reduced compared to the control genotype (*Figure 5E*). Upon further investigation of sleep bout duration, we found that the differences in sleep patterns between genotypes could be explained by the inability of *sxc*$^{C941Y}$ flies to maintain longer sleep bouts. *sxc*$^{WT}$ flies experience significantly more sleep bouts longer than 2 hr (median ± interquartile range 2.0 ± 1.0 bouts/24 hr) relative to *sxc*$^{C941Y}$ flies (1 ± 1.3 bouts). This aspect of sleep is also rescued by knocking out *Oga*, with *sxc*$^{C941Y}$;*Oga*$^{KO}$ flies no longer presenting with a significant decrease in number of sleep bouts longer than 2 hr (1.7 ± 1.3 bouts/24 hr) (*Figure 5G*). Accompanying decreased sleep bout duration, *sxc*$^{C941Y}$ and *sxc*$^{C941Y}$;*Oga*$^{KO}$ flies present with significantly more frequent sleep bouts during the day (mean ± standard deviation 13 ± 3 and 14 ± 5 bouts, respectively) and at night (14 ± 5 and 13 ± 5 bouts, respectively), compared to the control genotype (day: 11 ± 4 and night: 9 ± 4 bouts, respectively) (*Figure 5F*). These results indicate that the sleep defects in *sxc*$^{C941Y}$ flies are only partially rescued by elevating global O-GlcNAcylation, with the modest rescue of sleep bout duration seen upon loss of OGA fully rescuing total sleep, in part due to sleep frequency remaining unaltered and above the control genotype levels.

To dissect developmental from non-developmental contributions to this sleep phenotype, we investigated whether elevating O-GlcNAcylation only in adulthood could rescue the sleep phenotype observed in *sxc*$^{C941Y}$ flies. Adult *sxc*$^{C941Y}$ flies were fed 3 mM TMG for 72 hr prior to and during activity monitoring. In this condition, OGT-CDG flies no longer presented with decreased overall sleep duration (*Figure 6A*). This may be explained by differences in fly food used during this assay, to accommodate the addition of TMG. However, other aspects of sleep remained disrupted in OGT-CDG flies. Both mean sleep duration (median ± interquartile range, day: 22 ± 10 min, night: 44 ± 26 min) and daily number of sleep bouts longer than 2 hr (median ± interquartile range, 1 ± 1 bouts/24 hr) remained significantly reduced compared to the *sxc*$^{WT}$ control (day: 28 ± 13 min, night: 50 ± 43, 1.3 ± 1.0 bouts/24 hr, respectively). Additionally, as in previous experiments, *sxc*$^{C941Y}$ flies presented with significantly more sleep bouts throughout the day (mean ± standard deviation, day: 17 ± 4 bouts, night: 15 ± 5 bouts) than the control genotype (day: 14 ± 4 bouts, night: 12 ± 4 bouts) (*Figure 6B–D*). Interestingly, these phenotypes were partially reversed by TMG feeding. Mean sleep bout duration in *sxc*$^{C941Y}$ flies fed TMG was no longer significantly different from the control genotype both during the day and at night (26 ± 13 min and 44 ± 39 min, respectively), nor was the number of sleep bouts longer than 2 hr (1.3 ± 1.0 bouts/24 hr). The number of sleep bouts in *sxc*$^{C941Y}$ flies fed TMG was also no longer significantly different than for the control genotype fed a vehicle control (day: 16 ± 3 bouts, night: 13 ± 4 bouts), although it remained non-significantly elevated relative to the control genotype. This rescue was not due to non-specific effects of TMG on feeding behaviour, such as aversion due to altered food taste, as neither genotype nor inclusion of TMG in food influenced total feeding (*Figure 6—figure supplement 1*). These results suggest that effects of OGT-CDG mutations may not be solely developmental, and that defective O-GlcNAc cycling in adulthood may be an important contributor to the pathogenesis of these mutations.

## Glial knockdown of *sxc*$^{C941Y}$ partially phenocopies *sxc*$^{C941Y}$ fragmented sleep

Sleep in *Drosophila* and in humans is regulated by multiple cell types. To determine which cell types require normal O-GlcNAcylation to regulate sleep, we decided to knock down *sxc* in neurons and glia as both have been extensively implicated in this process (*Shafer and Keene, 2021*). Upon neuronal knockdown of *sxc*, no difference in total sleep (median ± interquartile range, 474 ± 228 min/24 hr) (*Figure 7A*) nor mean sleep bout duration (median ± interquartile range, day: 17 ± 10 min, night: 23 ± 14 min) (*Figure 7B*) was observed relative to the GAL4 control (total sleep 510 ± 232 min, mean sleep bout duration day: 11 ± 9 min, night: 28 ± 17 min). However, neuronal knockdown of *sxc* significantly affects sleep bout number, during the daytime decreasing (mean ± standard deviation, 8 ± 5

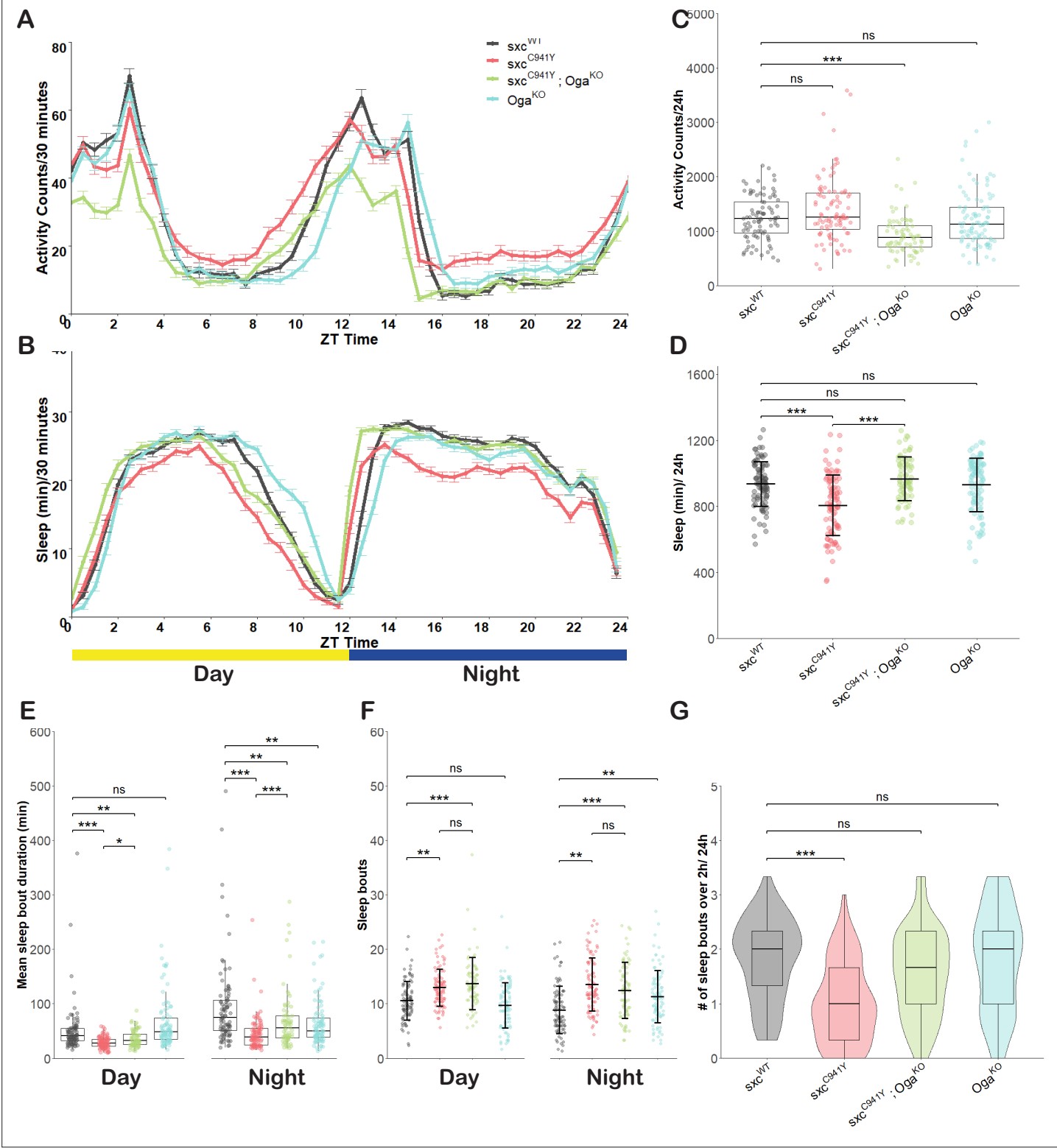

**Figure 5.** Fragmented sleep in a *Drosophila* model of OGT-CDG. (**A**) Activity profile (mean ± SEM of activity counts in 30 min bins) for *sxc^WT^* (n = 89), *sxc^C941Y^* (n = 94), *sxc^C941Y^;Oga^KO^* (n = 74), and *Oga^KO^* (n = 95) flies. (**B**) Sleep profile (mean ± SEM of sleep in 30 min bins) for genotypes as in (**A**). (**C–G**) Sleep parameters for genotypes in (**A**) and (**B**). (**C**) Total daily activity (median ± IQR) is significantly reduced in the *sxc^C941Y^;Oga^KO^* mutant strain relative to the control ($X^2(3)$ = 41.546, p<0.001, $p_{adj}$<0.001). (**D**) Total daily sleep (mean ± SD) is significantly reduced in *sxc^C941Y^* flies relative to the control genotype (F(3,348) = 18.34, p<0.001, $p_{adj}$<0.001), while both *sxc^C941Y^;Oga^KO^* and *Oga^KO^* flies do not have significantly altered total sleep ($p_{adj}$=0.58 and

*Figure 5 continued on next page*

*Figure 5 continued*

$p_{adj}$=0.99, respectively). (**E**) Mean sleep episode duration (median ± IQR) is significantly reduced in both $sxc^{C941Y}$ and $sxc^{C941Y}$;$Oga^{KO}$ flies relative to the control genotype during the day ($X^2(3)$ = 83.8, p<0.001, $p_{adj}$<0.001 and $p_{adj}$<0.01, respectively) and night ($X^2(3)$ = 52.0, p<0.001, $p_{adj}$<0.001 and $p_{adj}$<0.01, respectively). Mean sleep episode duration is significantly increased in $sxc^{C941Y}$;$Oga^{KO}$ flies compared to $sxc^{C941Y}$ flies both during the day and night ($p_{adj}$<0.05 and $p_{adj}$<0.001, respectively). (**F**) Daily number of sleep bouts (mean ± SD) is significantly elevated in both $sxc^{C941Y}$ and $sxc^{C941Y}$; $Oga^{KO}$ flies compared to the $sxc^{WT}$ control (F(3,696) = 20.31 p<0.001, $p_{adj}$<0.001 for both) while time of day had no significant effect on the number of sleep bouts (F(1,696) = 0.099, p=0.75). Post hoc analysis revealed that relative to the $sxc^{WT}$ control, the number of sleep bouts was significantly increased for $sxc^{C941Y}$ and $sxc^{C941Y}$;$Oga^{KO}$ flies both during the day ($p_{adj}$<0.01 and $p_{adj}$<0.001) and night ($p_{adj}$<0.001 and $p_{adj}$<0.001). (**G**) Daily number of sleep bouts longer than 2 hr (median ± IQR) is significantly lower in $sxc^{C941Y}$ flies than the control genotype ($X^2(3)$ = 49.623, p<0.001, $p_{adj}$<0.001). Individual points represent mean values of measurements conducted over 3 days, for unique flies. *p<0.05, **p<0.01, ***p<0.001.

The online version of this article includes the following source data for figure 5:

**Source data 1.** Raw DAM data and associated metadata (*Figure 5A–G*).

**Source data 2.** SCAMP output for individual flies including total sleep, total activity, mean sleep bout duration, and sleep bout number averaged over 3 days (*Figure 5C–F*).

bouts) relative to both the *elav* GAL4 (12 ± 4 bouts) and UAS (12 ± 4 bouts) control lines (*Figure 7C*). Conversely, knockdown of *sxc* in glial cells resulted in a significant decrease in total daily sleep (median ± interquartile range, 690 ± 230 min/24 hr), relative to a GAL4 (800 ± 170 min/24 hr), and UAS control (960 ± 120 min/24 hr) (*Figure 7A*). As in $sxc^{C941Y}$ flies, glial knockdown caused a decrease in the mean duration of sleep episodes (median ± interquartile range, day: 29 ± 26 min, night: 31 ± 19 min) compared with the control GAL4 (day: 38 ± 32 min, night: 46 ± 26 min) and UAS genotypes (day: 39 ± 24 min, night: 57 ± 40 min, *Figure 7B*); however, this effect is only significant during the night. Unlike in OGT-CDG flies, the number of sleep bouts was not significantly increased in flies expressing *sxc* RNAi in glial cells (mean ± standard deviation, day: 12 ± 6 bouts, night: 13 ± 5 bouts) relative to either control (GAL4: day: 12 ± 6 bouts, night: 11 ± 5 bouts, UAS: day: 12 ± 4 bouts, night: 10 ± 4 bouts; *Figure 7C*). Flies expressing *sxc* RNAi in glial cells presented with fewer sleep bouts longer than 2 hr (median ± interquartile range, 1.0 ± 1.3 bouts/24 hr), however, only relative to the UAS control group (2 ± 0.7 bouts/24 hr) and not the *repo* GAL4 line (1.3 ± 0.7 bouts/24 hr) (*Figure 7D*).

## Discussion

Many mutations in *OGT* causal in ID modelled previously do not result in a decrease in global O-Glc-NAcylation either in embryonic stem cells or patient derived fibroblasts, in many cases due to feedback mechanisms reducing OGA protein levels (*Willems et al., 2017*; *Pravata et al., 2019*; *Selvan et al., 2018*). The cysteine to tyrosine substitution modelled here is one of only two patient mutations which has been shown to reduce global O-GlcNAcylation when introduced in mammalian cells (*Pravata et al., 2020a*; *Omelková et al., 2023*). Here, we have shown that patient mutations affecting the catalytic domain of OGT result in decreased O-GlcNAcylation in adult flies, corroborating previous results (*Pravata et al., 2019*). Expanding upon these results, we have demonstrated that an OGT-CDG catalytic domain mutation can reduce O-GlcNAcylation throughout development, despite a compensatory increase in total *Dm*OGT protein. To assay O-GlcNAcylation levels, we used an antibody approach, yielding a coarse view of how OGT-CDG mutations affect the O-GlcNAcome, particularly given the known limitations and biases in using antibodies to detect this modification (*Thompson et al., 2018*). Though challenging, future research should expand on these results by quantitative analysis of changes to O-GlcNAcylation of specific protein substrates. When modelled in mouse embryonic stem cells, this mutation (C921Y in humans and mice) also results in an increase in OGT protein levels (*Omelková et al., 2023*). It is tempting to assert that increased OGT, as opposed to decreased OGA, is a homeostatic mechanism linked specifically to this mutation. However, in the fly, increased *Dm*OGT protein levels appear to be a response commonly associated with decreased OGT catalytic activity, demonstrated here by the $sxc^{K872M}$ mutant stain and in previous work (*Pravata et al., 2019*). While this increase in *Dm*OGT protein levels was not explored in further detail, some evidence exists for post-transcriptional regulation of *sxc* expression through alternate splicing (*Ashton-Beaucage et al., 2010*), a mechanism known be involved in the control of *OGT* and *OGA* expression and O-GlcNAc homeostasis in mammalian cells (*Tan et al., 2020*). We have also shown that reduced catalytic activity of *Dm*OGT as a result of modelling a patient mutation in *sxc* can phenocopy rational mutagenesis of a

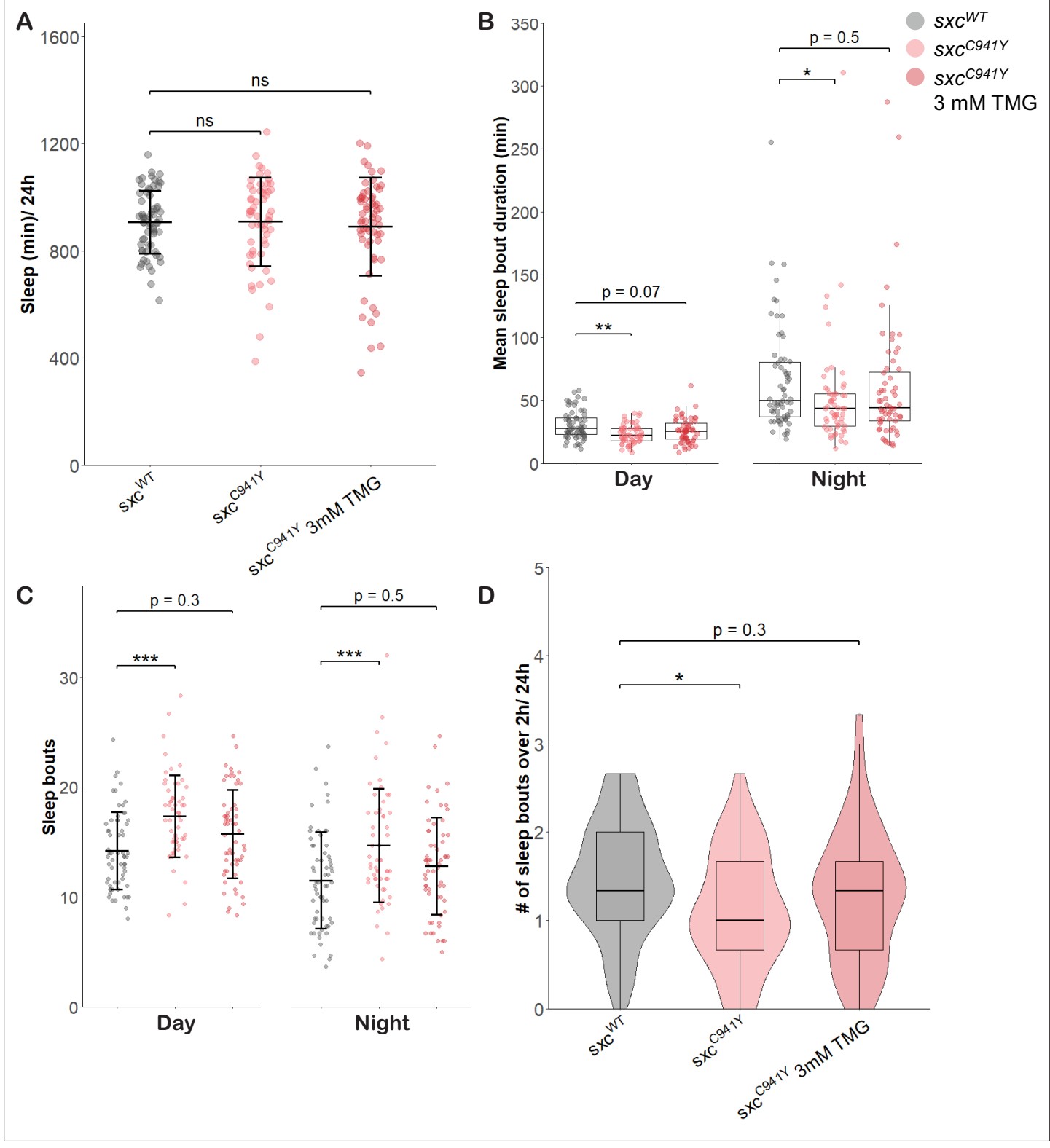

**Figure 6.** Thiamet G partially rescues sleep in adult $sxc^{C941Y}$ flies. Sleep parameters for $sxc^{WT}$ (n = 63), $sxc^{C941Y}$ (n = 57), and $sxc^{C941Y}$ flies fed 3 mM Thiamet G (TMG) (n = 60).(**A**) Total sleep (mean ± SD) is not significantly different between groups (F(2,177) = 0.249, p=0.78). (**B**) Mean sleep bout duration (median ± IQR) significantly differs between groups during the day ($X^2$(2) = 12.854, p<0.01) and night ($X^2$(2) = 6.5027, p<0.05), though it is only significantly reduced for $sxc^{C941Y}$ flies fed the vehicle control ($p_{adj}$=0.001 and $p_{adj}$<0.05) and not TMG ($p_{adj}$=0.07, $p_{adj}$=0.48). (**C**) Daily number of sleep bouts is significantly different across groups (F(2,354) = 16.775, p<0.001) and times of day (F(2,354) = 37.446, p<0.001). Post hoc analysis reveals that compared

*Figure 6 continued on next page*

*Figure 6 continued*

to the $sxc^{WT}$ genotype, $sxc^{C941Y}$ flies fed a vehicle control present with significantly more sleep bouts both during the day and at night ($p_{adj}<0.001$ for both) while the same genotype fed TMG supplemented food did not present with a significantly different number of sleep bouts ($p_{adj}=0.33$ and $p_{adj}=0.52$). (**D**) The number of sleep bouts longer than 2 hr is significantly reduced in $sxc^{C941Y}$ flies fed a vehicle control, but not TMG ($X^2(2) = 8.2491$, $p<0.05$, $p_{adj}<0.05$ and $p_{adj}=0.3$, respectively). Individual points represent mean values of measurements conducted over 3 days, for unique flies. *$p<0.05$, **$p<0.01$, ***$p<0.001$.

The online version of this article includes the following source data and figure supplement(s) for figure 6:

**Source data 1.** Raw DAM data and associated metadata (*Figure 6A–D*).

**Source data 2.** SCAMP output for individual flies including total sleep, mean sleep bout duration, and sleep bout number averaged over 3 days (*Figure 6A–C*).

**Source data 3.** OD625 of lysed flies fed food with Blue Dye no 1, either with or without Thiamet G.

**Figure supplement 1.** Thiamet G supplementation does not alter total feeding in adult flies.

---

key catalytic residue ($sxc^{H537A}$) causing the growth of ectopic scutellar bristles (*Mariappa et al., 2018*). Also known as macrochaetae, the development of these sensory cells is well studied, particularly in the context of cell fate determination by lateral inhibition through Notch signalling (*Parks et al., 1997*), providing a tractable system for the understanding of the impacts of hypo-O-GlcNAcylation on cell fate determination. With reports of Notch signalling requiring appropriate O-GlcNAcylation (*Chen et al., 2021*), this phenotype presents an interesting system to research the contribution of the Notch signalling pathway to ID.

A key question regarding OGT-CDG is whether therapeutic approaches targeting OGA can raise O-GlcNAcylation and potentially ameliorate symptoms in this disorder, as previously proposed (*Pravata et al., 2020b*). Here, we show that normal global O-GlcNAcylation levels can be restored in $sxc^{C941Y}$ adult flies through knockout out of *Oga*. Previous research has demonstrated that *Oga^{KO}* alleles can rescue phenotypes associated with reduced O-GlcNAcylation (*Fenckova et al., 2022*); however, here we present the first direct evidence that global O-GlcNAcylation can exceed control levels in *Dm*OGT hypomorphic flies in the absence of OGA. While we did not see a concomitant rescue of the scutellar bristle phenotype seen in $sxc^{C941Y}$ flies, this may occur as a consequence of unique kinetics of O-GlcNAcylation and removal of O-GlcNAc on various *Dm*OGT substrates, that is, the dysregulation of the ratio of stoichiometries of modification of specific substrates may be exacerbated in the absence of OGA. Previous research has demonstrated this may occur in mammalian cells, with some O-GlcNAcylated proteins not affected by OGA inhibition in cancer cells (*Hahne et al., 2013*; *Li et al., 2019*). Alternatively, it may be that both the addition and timely removal of O-GlcNAc from specific proteins is required for normal scutellar bristle development.

Complete ablation of *Oga* expression is a blunt approach, elevating O-GlcNAcylation levels beyond those seen in the control genotype and is not a feasible therapeutic approach. Pharmacological approaches to inhibit OGA offer more precise control over the degree of O-GlcNAcase activity and are being actively pursued as potential treatments for neurodegenerative disorders (*Bartolomé-Nebreda et al., 2021*). Our experiments suggest that the potent OGA inhibitor TMG can be used to rescue global O-GlcNAcylation levels in $sxc^{C941Y}$ flies to those of a genetic background control, at various stages of development. Interestingly, rescuing O-GlcNAcylation levels through OGA inhibition also restored OGT levels in $sxc^{C941Y}$ flies. However, there is a clear difference in the pattern of O-GlcNAcylation visualised by immunoblotting in adult $sxc^{C941Y}$ flies fed TMG relative to the control genotype. This incomplete rescue of O-GlcNAcylation may be consequential in phenotypic rescue. Additionally, a paradoxical effect was seen upon feeding higher doses of TMG. Because inhibition of OGA appears to reduce protein levels of OGT, global O-GlcNAc levels were not rescued at higher levels of TMG. However, specific immunoreactive bands appeared to maintain elevated levels of O-GlcNAc. This potentially indicates that mechanisms controlling OGT expression in response to O-GlcNAcylation levels are particularly sensitive to OGA activity, lowering OGT protein levels prior to O-GlcNAcylation stoichiometry rising on some OGT substrates.

Previous research has shown that alleles encoding hypomorphic variants of *Dm*OGT result in overgrowth at the NMJ, and that TPR domain mutations modelling those seen in patients result in a similar phenotype (*Fenckova et al., 2022*). However, here, we observe the opposite effect, with both $sxc^{C941Y}$ and $sxc^{K872M}$ larvae presenting with smaller NMJs. This discrepancy is unlikely to be caused

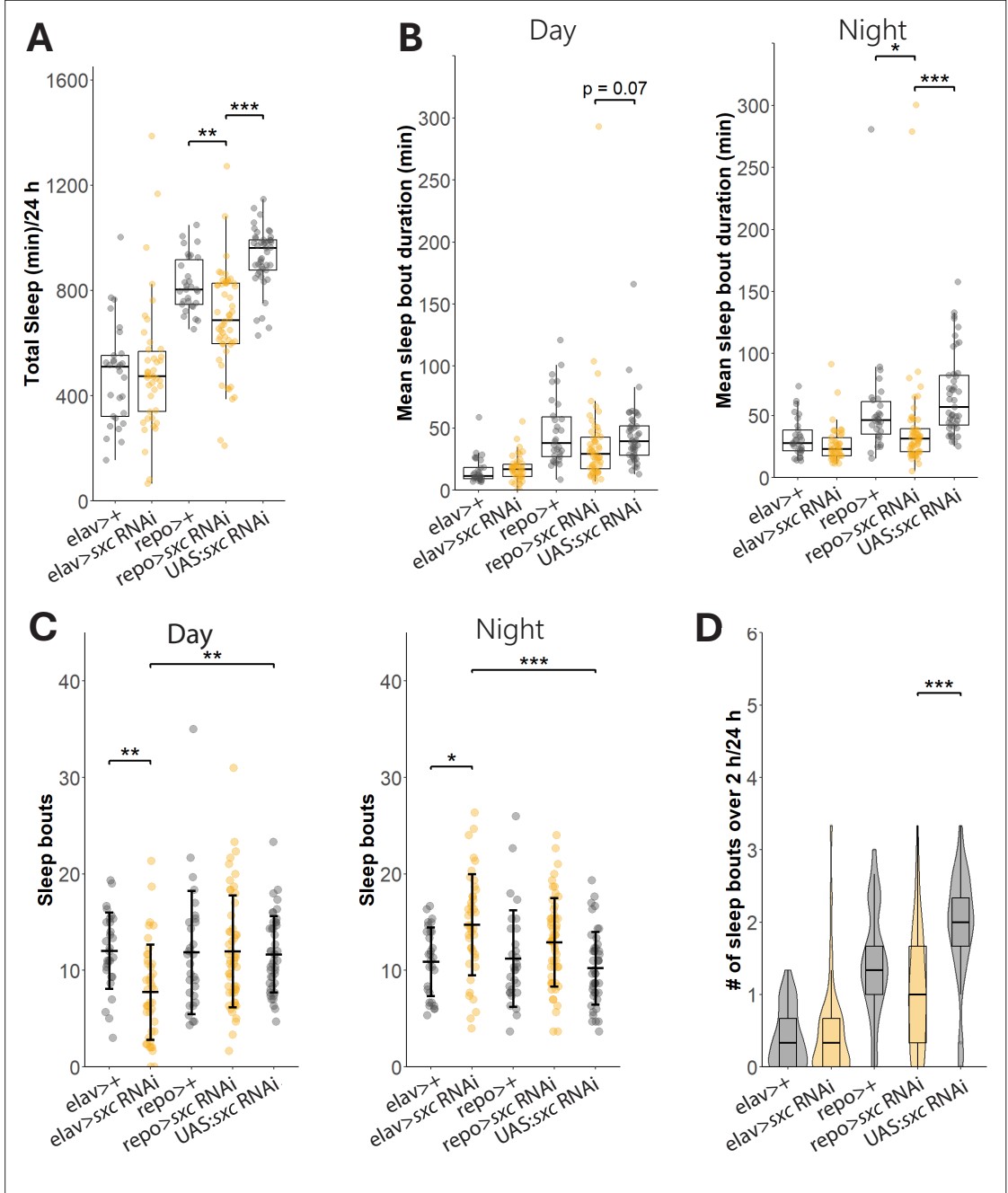

**Figure 7.** Glial knockdown of *sxc* partially phenocopies *sxc*[C941Y] sleep defects. Quantification of sleep parameters in flies expressing *sxc* RNAi under the control of either the neuronal *elav* promoter (n = 42) or the glial *repo* promoter (n = 54). (**A**) Total sleep is significantly ($\chi^2(4)$ = 105.98, p<0.001) reduced by *sxc* knockdown in glial cell, compared to both to the *repo* >GAL4 (n = 30, $p_{adj}$<0.01) and UAS (n = 45, $p_{adj}$<0.001) control lines. (**B**) Mean sleep episode duration is only significantly reduced in *repo>sxc* RNAi flies during the night, compared to both the UAS and GAL4 control lines ($\chi^2(4)$ = 67.073, p<0.001, $p_{adj}$<0.001 and $p_{adj}$<0.05, respectively). (**C**) Daily number of sleep episodes is not significantly affected by genotype (F(4, 127) = 1.368, p=0.24); however, there is a significant effect of time of day on sleep (F(4, 127) = 5.656, p<0.05). There is also a significant interaction between genotype and time of day with regards to number of sleep bouts (F(4, 127) = 10.747, p<0.001), which can be explained by a significant decrease in number of sleep bouts during the day in neuronal *sxc* knockdown flies compared to both the *elav*>GAL4 (n = 29, $p_{adj}$<0.01) and UAS ($p_{adj}$<0.01) control lines. Conversely, neuronal *sxc* knockdown flies present with significantly more sleep bouts during the night, compared to these control lines ($p_{adj}$<0.05 and $p_{adj}$<0.01, respectively). (**D**) The daily number of sleep episodes longer than 2 hr is also reduced with glial *sxc* knockdown ($\chi^2(4)$ = 81.215, p<0.001), but only compared to the UAS control line ($p_{adj}$<0.001) and not the GAL4 control line ($p_{adj}$=0.3). *p<0.05, **p<0.01, ***p<0.001.

The online version of this article includes the following source data for figure 7:

**Source data 1.** Raw DAM data and associated metadata (**Figure 7A–H**).

*Figure 7 continued on next page*

Figure 7 continued

**Source data 2.** SCAMP output for individual flies including total sleep, mean sleep bout duration, and sleep bout number averaged over 3 days (*Figure 7A–C*).

by allele-specific effects as this hypothesis was tested by assaying NMJ parameters in one of the hypomorphic mutants previously described and finding that this mutation ($sxc^{H596F}$) also results in stunted growth at the NMJ. While we utilised different markers to count boutons than in previous research (*Fenckova et al., 2022*), discrepancies in area and length of NMJs cannot be explained in this manner. Nonetheless, as in previous research on the effects of reduced OGT catalytic activity on NMJ morphology, knocking out *Oga* can partially rescue phenotypes at this type of synapse (*Fenckova et al., 2022*), while a lack of *Oga* on its own does not affect NMJ morphology. Previous research on the role of *Oga* in axonal outgrowth at the larval NMJ is contradictory, with one report indicating a lack of functional OGA protein increases NMJ morphological parameters such as area and another suggesting no effect on NMJ growth in the absence of OGA (*Muha et al., 2020Fenckova et al., 2022*). Additionally, we also show that overall area of the NMJ can be elevated by overexpressing wild type *sxc* in neurons, in a *Dm*OGT catalytically dead background ($sxc^{K872M}$). Surprisingly, other parameters (such as bouton number and NMJ length) were not elevated by such overexpression, neither when targeting neurons nor muscle cells. This phenotype can also be partially rescued through pharmacological means, through the use of TMG. This could serve as proof of principle that pharmacological intervention in OGT-CDG is possible. This is not an immediately obvious conclusion as it is possible that beyond the stoichiometry of the modification on individual substrates the timing of addition and removal of O-GlcNAcylation could be important for synaptic development and function. Together, these findings suggest that the NMJ phenotype caused by loss of normal O-GlcNAcylation is complex, and that potentially alternate mechanisms impair bouton addition (which may also affect NMJ length) and overall NMJ area, as the latter is more amenable to rescue across various approaches.

We also demonstrate a novel behavioural effect resulting from catalytic domain mutations in *sxc*. Normal O-GlcNAc cycling is required for maintenance of sleep in adult flies, and reduced *Dm*OGT catalytic activity as a result of an OGT-CDG mutation results in shorter, more frequent sleep episodes. Previously, a patient with this condition was reported to suffer from sleep disturbances characterised by abnormal EEG during sleep and insomnia (*Selvan et al., 2018*). This is particularly relevant given that normal sleep is required for multiple cognitive processes, such as memory formation, both in flies and humans (*Ly et al., 2018*; *Berry et al., 2015*; *Rasch and Born, 2013*). Encouragingly, this phenotype can be partially rescued by knocking out OGA or by pharmacologically elevating O-GlcNAcylation in adult flies. However, it is important to note that the phenotype was not consistently detected across varying experimental conditions. Specifically, when sleep was assayed in OGT-CDG flies fed an alternate food type (to accommodate the addition of TMG) total sleep was not reduced in $sxc^{C941Y}$ flies, despite bout duration and frequency being altered as for assays using standard *Drosophila* food. While not directly comparable, this may suggest that loss of OGT catalytic activity interacts with environmental factors such as dietary composition. Further investigation revealed that knocking down *sxc* in glial cells can result in a similar phenotype as the OGT-CDG mutation. While this does not rule out the involvement of neuronal cell types in this phenotype, the results presented here provide the first evidence for the potential involvement of non-neuronal cell types in OGT-CDG. Overall, these results could have important implications for our understanding of OGT-CDG, providing the first evidence that suggests that the disorder is not purely developmental and may be amenable to therapeutic approaches at later stages of life.

# Materials and methods

**Key resources table**

| Reagent type (species) or resource | Designation | Source or reference | Identifiers | Additional information |
|---|---|---|---|---|
| Gene (*Drosophila melanogaster*) | *sxc* | FlyBase | FLYB: FBgn0261403 | |
| Gene (*D. melanogaster*) | *Oga* | FlyBase | FLYB: FBgn0261403 | |
| Strain, strain background (*D. melanogaster*) | w[1118] (VDRC60000) | Vienna *Drosophila* Resource Center | VDRC60000 | |
| Genetic reagent (*D. melanogaster*) | vas-Cas9 | Bloomington *Drosophila* Stock Center | BL51323 | |
| Genetic reagent (*D. melanogaster*) | *sxc[K872M]* | **Mariappa et al., 2018** | FLYB: FBal0340183 | |
| Genetic reagent (*D. melanogaster*) | *Oga[KO]* | **Muha et al., 2020** | FLYB: FBal0361594 | |
| Genetic reagent (*D. melanogaster*) | *sxc[N595K]* | **Pravata et al., 2019** | FLYB: FBal0352246 | |
| Genetic reagent (*D. melanogaster*) | *sxc[H596F]* | **Fenckova et al., 2022** | FLYB: FBal0375027 | |
| Genetic reagent (*D. melanogaster*) | *sxc* RNAi | Vienna *Drosophila* Resource Center | VDRC110717 | |
| Genetic reagent (*D. melanogaster*) | elav-GAL4 | Bloomington *Drosophila* Stock Center | BL8765 | |
| Genetic reagent (*D. melanogaster*) | repo-GAL4 | Kind gift from Leeanne McGurk | | |
| Genetic reagent (*D. melanogaster*) | mhc- GAL4 | Bloomington *Drosophila* Stock Center | BDSC_55133 | |
| Genetic reagent (*D. melanogaster*) | elavL3-GAL4 | Kind gift from Leeanne McGurk | RRID:BDSC_8760 | |
| Genetic reagent (*D. melanogaster*) | UAS:sxc-HA | **Mariappa et al., 2015** | | |
| Antibody | RL2, anti-O-GlcNAc (mouse monoclonal) | Novus | Cat# NB300-524 | 1:1000 |
| Antibody | Anti-OGT (rabbit polyclonal) | Abcam | Cat# ab-96718 | 1:1000 |
| Antibody | Anti-Actin (rabbit polyclonal) | Sigma | Cat# A2066 | 1:5000 |
| Antibody | Anti-Discs Large 1 (mouse polyclonal) | DSHB | RRID:AB_528203 | 1:25 |
| Antibody | Anti-HRP conjugated to Alexa Fluor 647 (goat polyclonal) | Jackson ImmunoResearch | RRID:AB_528203 | 1:400 |
| Antibody | Anti-rabbit IgG 680 infrared conjugated (donkey polyclonal) | LI-COR | RRID:AB_2716687 | 1:10,000 |
| Antibody | Anti-mouse IgG 800 infrared conjugated (goat polyclonal) | LI-COR | RRID:AB_2687825 | 1:10,000 |
| Antibody | Anti-mouse conjugated to Alexa Fluor 488 (donkey polyclonal) | Molecular Probes | A21202 | 1:400 |
| Recombinant DNA reagent | pCFD3-dU63gRNA | Addgene | | |

*Continued on next page*

*Continued*

| Reagent type (species) or resource | Designation | Source or reference | Identifiers | Additional information |
|---|---|---|---|---|
| Peptide, recombinant protein | *CpOGA* (GST tagged) | *Rao et al., 2006* | | Produced in-house |
| Chemical compound, drug | Thiamet G | SantaCruz | Cat# sc-224307 | |
| Chemical compound, drug | NuPage LDS | Thermo Fisher | Cat# NP0007 | |
| Chemical compound, drug | Dako mounting media | Agilent | Cat# S302380-2 | |
| Chemical compound, drug | GlcNAcstatin G | *Dorfmueller et al., 2010* | | Produced in-house |
| Chemical compound, drug | Blue dye no. 1 | Sigma | 861146 | |
| Software, algorithm | Image Studio Lite | https://www.licor.com/bio/image-studio/ | | |
| Software, algorithm | DAMFileScan113 | https://www.trikinetics.com/ | | |
| Software, algorithm | Python 3 | https://www.python.org/ | | |
| Software, algorithm | R (4.0.3) | https://www.r-project.org/ | | |
| Software, algorithm | ImageJ-FIJI | https://imagej.net/software/fiji/ | | |
| Software, algorithm | SCAMP | *Donelson et al., 2012* | | |
| Software, algorithm | Rethomics | *Geissmann et al., 2019* | | |
| Software, algorithm | Drosophila_NMJ_Morphometrics | *Nijhof et al., 2016* | | |
| Software, algorithm | WBplotProfile | This paper, *Czajewski, 2024* | | Code can be obtained from: https://github.com/IgnacyCz/WBplotProfile |
| Other | Confocal microscope | Zeiss | 710 | Section 'Neuromuscular junction immunohistochemistry' |
| Other | Confocal microscope | Zeiss | 980 | Section 'Neuromuscular junction immunohistochemistry' (Neuronal overexpression of *sxc* partially rescues neuromuscular junction defects caused by loss of OGT catalytic activity) |
| Other | Dissection microscope | Motic | SMZ-161 | Section 'Scutellar bristle assay' |
| Other | DAM2 monitor | trikinetics | https://trikinetics.com/ | Section '*Drosophila* activity monitor' |

## CRIPSR-Cas9 mutagenesis

The gRNA sequence for generating the sxc C941Y flies was selected using the online tool Crispr.mit. edu. The optimal gRNA sequence was included in annealing oligos including overhangs compatible with cloning into the pCFD3-dU63gRNA plasmid previously cut with *Bpi*I restriction enzyme. A 2 kb repair template for the region was generated from *Drosophila* Schneider 2 cell genomic DNA by PCR using GoTaq G2 Polymerase. The PCR product was cloned as a blunt product into the pTOPO-Blunt plasmid. Mutations were introduced into the template to include the C941Y mutation as well as silent mutations to remove the gRNA recognition sequence. This was carried out using the QuikChange kit from Stratagene and confirmed by DNA sequencing. The mutations removed the restriction site BseMI which is present in the gRNA sequence. *sxc*^C941Y^ mutant flies were generated by microinjection of vas-Cas9 embryos (BL51323) (Rainbow Transgenic Flies, Inc) with CRISPR reagents generated in-house,

**Table 1.** List of oligonucleotides used in *CRISPR-Cas9 mutagenesis* section.

| Designation | Sequence |
| --- | --- |
| gRNA annealing oligonucleotide (forward) | GTCGCTTGATACTCCTTTATGCAA |
| gRNA annealing oligonucleotide (reverse) | AAACTTGCATAAAGGAGTATCAAG |
| Repair template cloning primer (forward) | aaaGGATCCTTTCGACACAAAATCAGTCGAGAGTCTG |
| Repair template cloning primer (reverse) | aaaGCGGCCGCGGTAGCCAGCTGAGAGGCAGCCAC |
| Repair template mutagenesis primer 1 (silent) (forward) | GGGGTCAATTAGCTGATATATGTCTTGATACgCCgcTgTcgAATGGGCATACAACATCTATGGACGTTTTG |
| Repair template mutagenesis primer 1 (silent) (reverse) | CAAAACGTCCATAGATGTTGTATGCCCATTcgAcAgcGGcGTATCAAGACATATATCAGCTAATTGACCCC |
| Repair template mutagenesis primer 2 (C941Y mutation) (forward) | GTCAATTAGCTGATATATacCTTGATACGCCGCTGTgtAATGGGCATACAACATCTATG |
| Repair template mutagenesis primer 2 (C941Y mutation) (reverse) | CATAGATGTTGTATGCCCATTacACAGCGGCGTATCAAGgtATATATCAGCTAATTGAC |

backcrossed to a w[1118] (VDRC60000) background and the mutated chromosome was balanced over Curly of Oster (CyO). Diagnostic digests were carried out on the resulting flies to first confirm the loss of the restriction site followed by sequencing of the PCR product. The correctness of the mutation was also confirmed through sequencing of the full-length *sxc* mRNA. Sequences of the oligonucleotides used here are listed in *Table 1*.

## Fly stocks and maintenance

Stocks were maintained on a 12:12 light dark cycle at 25°C on Nutri-Fly Bloomington Formulation fly food. Previously described *sxc*[K872M] (*Mariappa et al., 2018*), *sxc*[N595K] (*Pravata et al., 2019*), and *sxc*[H596F] (*Fenckova et al., 2022*) mutant flies (2018) were used. Previously described *Oga*[KO] flies (*Muha et al., 2020*) were used to generate *sxc*[C941Y];*Oga*[KO] and *sxc*[K872M];*Oga*[KO] stocks. The homozygous lethal *sxc*[K872M] chromosome was balanced over a CyO chromosome carrying a GFP reporter (CyO, P{ActG-FP.w[-]}CC2, BL9325). An isogenic w[1118] (VRDC60000) background strain was used as a control genetic background. To overexpress wild type *sxc* in an *sxc*[K872M] background, *sxc*[K872M]/CyO(GFP);mhc-Gal4 (generated using the w[*]; P{w[+mC]=Mhc-GAL4.K}2/TM3, Sb[1] line; BL55133) or *sxc*[K872M]/CyO(GFP);elavL3-Gal4 (generated using the P{GAL4-elav.L}CG16779[3] line; RRID:BDSC_8760) were crossed with *sxc*[K872M]/CyO(GFP);UAS:sxc-HA (generated using the previously described UAS:sxc-HA line; *Mariappa et al., 2015*). To knock down *sxc* in neurons or glia, the *sxc* RNAi line VDRC110717 was used and 10 of either virgin VDRC110717 or VRDC60000 females were crossed with five P{w[+mC]=GAL4elav.L}2/CyO (BL8765), p{w[mC]:repo-GAL4}/TM6b (kind gift from Leeanne McGurk), or VRDC60000 males and allowed to lay embryos for 5 days.

## *Drosophila* tissue lysis and western blotting

For immunoblotting of adult head lysates, flies raised as described previously were anaesthetised with $CO_2$ and an equal number of 3–5-day-old male and female flies were snap frozen in liquid nitrogen. Heads were then severed from bodies by vortexing flies twice and collected using a paintbrush. To collect larval and embryonic lysates, homozygous 3–5-day-old females and males were allowed to lay embryos for 4 hr and 2 hr, respectively, on apple juice agar plates supplemented with yeast paste. For recessive lethal lines, heterozygous parents were crossed in the same manner. Embryos were collected 14 hr later and snap frozen on dry ice. For recessive lethal genotypes, homozygous embryos were collected based on the absence of a GFP fluorescent CyO balancer chromosome. For larval tissues, 24 hr after embryo collection, *sxc* mutant homozygous first-instar larvae were collected into vials containing Nutri-Fly Bloomington Formulation fly food at a density of 25 larvae per vial and aged to the wandering third-instar stage, when they were snap frozen on dry ice. For experiments in

which specificity of the O-GlcNAc antibody was tested by prior incubation with *Clostridium perfringens OGA CpOGA,* heads were lysed in modified RIPA buffer to accommodate the pH optimum of *Cp*OGA (*Rao et al., 2006*) (150 mM NaCl, 1% NP-40, 0.5% sodium deoxycholate, 0.1% SDS, 25 mM citric acid pH 5.5) supplemented with a protease inhibitor cocktail (1 M benzamidine, 0.2 mM PMSF, 5 mM leupeptin). To validate specificity of O-GlcNAc detection, lysates were split with one group incubated with 2.5 µM GST tagged *Cp*OGA to remove O-GlcNAc while the experimental group was incubated with 1 µM GlcNAcstatin G. Lysates were then incubated for 2 hr at room temperature, agitated at 300 RPM using a thermomixer (Eppendorf thermomixer comfort). The reaction was stopped by heating to 95°C with NuPAGE LDS Sample Buffer with 50 mM TCEP to a 1× concentration. Otherwise, collected tissues were lysed in 50 mM Tris- HCl (pH 8.0), 150 mM NaCl, 1% Triton-X 100, 4 mM sodium pyrophosphate, 5 mM NaF, 2 mM sodium orthovanadate, 1 mM EDTA, supplemented 1:100 with a protease inhibitor cocktail (1 M benzamidine, 0.2 mM PMSF, 5 mM leupeptin) and 1.5× NuPAGE LDS Sample Buffer with 50 mM TCEP. Protein concentration was estimated using a Pierce 660 assay (Thermo Scientific) supplemented with ionic detergent compatibility reagent (Thermo Scientific). 30 µg of protein per group were separated by gel electrophoresis (NuPage 4–12% Bis-Tris, Invitrogen) and transferred onto a nitrocellulose membrane (Amersham Protran 0.2 µm). Membranes were developed with the following primary antibodies: mouse anti-O-GlcNAc (RL2, 1:1000, Novus), rabbit anti-OGT (1:1000, Abcam, ab-96718), and rabbit anti-actin (1:5000, Sigma, A2066) and the following secondary antibodies: goat anti-mouse IgG 800 and donkey anti-rabbit IgG 680 infrared dye conjugated secondary antibodies (LI-COR, 1: 10,000). Western blots were analysed using Image Studio Lite.

## Thiamet G feeding

Thiamet G (SantaCruz, sc-224307) was dissolved in PBS to a stock concentration of 100 mM. This stock was mixed with *Drosophila* instant food (Flystuff Nutri-Fly Food, Instant Formulation) to appropriate concentrations, to avoid heating Thiamet G. For experiments with adult flies, 1–3-day-old flies (males and females in equal proportion) were placed on food for 72 hr prior to snap freezing in liquid nitrogen. For larval feeding experiments, ten 0–3-day-old females were crossed with four males and allowed to lay embryos for 2 days. Wandering third-instar larvae were snap frozen on dry ice and lysed.

Effects of the addition of Thiamet G to food on adult feeding behaviour were assayed similarly to *Wong et al., 2009*. To age match flies, freshly eclosed adults were placed on standard food as described in 'Fly stocks and maintenance' for 2 days. Males were then transferred to vials with 1% agarose for 18 hr, to starve flies and later induce feeding. After starvation, flies were placed in either vehicle control vials (0.86% agarose, 5 mM sucrose, 3% PBS, 0.2% blue dye no. 1 [Sigma 861146]) or Thiamet G containing vials (0.86% agarose, 5 mM sucrose, 3% PBS, 3 mM Thiamet G, 0.2% blue dye no. 1) for 30 min. Ten flies per replicate were then ground with a pestle in 50 µL of water and centrifuged at 17,000 RCF for 15 min to remove debris. 35 µL of supernatant was then collected and absorbance at 625 nm was measured (NanoDrop One, Thermo Fisher Scientific).

## NMJ immunohistochemistry

The NMJ assay was performed as in *Nijhof et al., 2016*. Larvae for this assay were obtained as described above. Male wandering third-instar larvae were dissected using the 'open book' technique *Brent et al., 2009* followed by immediate fixation in 3.7% paraformaldehyde in phosphate buffered saline (pH 7.5) (PBS) for 25 min. Fixed larvae were either stored in PBS at 4°C for up to 48 hr or immediately processed further. Larval preparations were blocked using 5% normal donkey serum (NDS) in PBS and Triton-X (0.3%, PBST) for 2 hr at room temperature, followed by immunostaining using mouse anti-Discs Large 1 (1:25, Developmental Studies Hybridoma Bank, RRID:AB_528203) and goat anti-HRP conjugated to Alexa Fluor 647 (1:400, Jackson ImmunoResearch, RRID:AB_2338967) in 5% NDS PBST overnight at 4°C. Sections were washed four times for 10 min in PBST (0.5%), followed by 4 hr incubation with donkey anti-mouse Alexa Fluor 488 (1:400) in 5% NDS PBST at room temperature. Sections were washed as for primary antibodies, rinsed in PBS, and mounted using Dako Fluorescence Mounting Medium (Agilent). Images of type 1b NMJs of muscle 4 were obtained using either a Zeiss 710 or 980 confocal microscope using a ×10 objective (EC Plan Neofluar 0.3) (voxel size: 0.69 × 0.69 × 6.22 µm) for muscle area measurements and using a ×63 objective (Plan Apochromat 1.4 oil) to image individual junctions (voxel size: 0.196 × 0.196 × 0.91 µm). Image size for the former was 2048 × 2048 pixels and 688 × 688 for the latter. Both channels were acquired simultaneously. NMJ parameters

were scored using a semi-automated macro by a researcher blind to the conditions (Neuromuscular Junction Morphometrics; *Nijhof et al., 2016*) with poorly annotated or damaged NMJs excluded from further analysis. Muscle area was manually measured using the polygon selection tool in ImageJ. Statistical analysis was performed on mean values for individual larvae for which three or more NMJs were accurately annotated.

### *Drosophila* activity monitor

*Drosophila* activity was recorded using Trikinetics DAM2 monitors. 1–3-day-old male flies were used for all experiments. Briefly, male flies were anaesthetised using $CO_2$ and placed in DAM vials with Nutri-Fly Bloomington Formulation food. Experiments were performed at 25°C on a 12 hr:12 hr light:dark cycle, data were recorded for 3 days, after 2 days of acclimatisation. For TMG rescue experiments, food was prepared as described in 'Thiamet G feeding' and data were recorded 72 hr after placing flies on supplemented food. Data were pre-processed using DAMFileScan113 software and Sleep and Circadian Analysis MATLAB Program (SCAMP) (*Donelson et al., 2012*). Flies that ceased to move during the experimental window were presumed dead and excluded. For analysis of number of bouts longer than 2 hr, the raw output from the DAM system was analysed in R using the Rethomics packages (*Geissmann et al., 2019*).

### Scutellar bristle assay

To assay scutellar bristle number, 8–10 young homozygous virgin females were mated with three males and allowed to lay embryos for 3 days to prevent overcrowding of larvae. Eclosed offspring were immobilised using $CO_2$ and scutellar bristles were counted using a Motic SMZ-161 microscope.

### Western blot intensity profile

Intensity of O-GlcNAc immunoreactivity was calibrated to estimated molecular weight plotted using custom Python code (available at GitHub, copy archived at *Czajewski, 2024*). Briefly, images were imported using the PIL library (*Umesh, 2012*), converted to NumPy arrays (*Harris et al., 2020*), and molecular weight markers were identified as intensity peaks in a user-defined x-coordinate column of pixels. The SciPy (*Virtanen et al., 2020*) library was then used to fit a curve to identified molecular weight markers to infer molecular weights at y-pixel coordinates. This was then used to calibrate the x-axis for plotting (using the matplotlib library; *Hunter, 2007*) the relative intensity of immunolabelling across genotypes and conditions based on user defined x pixel coordinates defining protein lanes, normalised to loading controls.

### Statistical analyses

All statistical analyses were performed in R (version 4.0.3). Data that satisfied assumptions regarding homoscedasticity and normality were analysed with a one-way ANOVA followed by Tukey's HSD with Bonferroni correction. Otherwise, data were analysed using a Kruskal–Wallis rank sum test followed by pairwise comparisons using Wilcoxon rank sum test with continuity correction and p value adjustment using the Bonferroni method. Sleep architecture phenotypes which were analysed separately for data collected during the day and night were analysed by two-way ANOVA (sleep bout number) or by a Kruskal–Wallis rank sum test performed separately on the two time periods (mean sleep bout duration), followed by post hoc testing as above. One outlier was removed from analysis ($sxc^{N595K}$ OGT and O-GlcNAcylation quantification), based on the criteria of falling more than 1.5 interquartile range beyond the 75 percentile. To balance the removal of this outlier, the minimum for this group was also removed.

### Materials availability statement

All materials are available upon request.

## Acknowledgements

This work was funded by a Wellcome Trust Investigator Award (110061), a Novo Nordisk Foundation Laureate award (NNF21OC0065969), and a Villum Fonden Investigator (00054496) to DMFvA, and a PhD studentship from the National Centre for the Replacement, Refinement and Reduction of Animals in Research (NC3Rs, award number T001682).

Supported in part by the Danish Research Institute of Translational Neuroscience – DANDRITE of the Nordic-EMBL Partnership for Molecular Medicine and Lundbeckfonden. We also thank Leeanne McGurk and Jens Januschke for their feedback as well as past and current members of our laboratory for their input, including Hannah Smith, Marta Murray, Veronica Pravata, and Conor Mitchell.

## Additional information

### Funding

| Funder | Grant reference number | Author |
| --- | --- | --- |
| Wellcome Trust | 110061 | Daan MF van Aalten |
| Novo Nordisk Foundation | NNF21OC0065969 | Daan MF van Aalten |
| National Centre for the Replacement Refinement and Reduction of Animals in Research | T001682 | Ignacy Czajewski |
| Villum Fonden | Villum Investigator 00054496 | Daan MF van Aalten |

The funders had no role in study design, data collection and interpretation, or the decision to submit the work for publication. For the purpose of Open Access, the authors have applied a CC BY public copyright license to any Author Accepted Manuscript version arising from this submission.

### Author contributions

Ignacy Czajewski, Conceptualization, Formal analysis, Validation, Investigation, Visualization, Methodology, Writing – original draft; Bijayalaxmi Swain, Jiawei Xu, Investigation; Laurin McDowall, Data curation, Investigation, Methodology; Andrew T Ferenbach, Investigation, Methodology, Writing - review and editing; Daan MF van Aalten, Conceptualization, Resources, Formal analysis, Supervision, Funding acquisition, Investigation, Project administration, Writing - review and editing

### Author ORCIDs

Ignacy Czajewski ⓘ https://orcid.org/0000-0002-1661-5806
Bijayalaxmi Swain ⓘ http://orcid.org/0000-0003-0885-0395
Daan MF van Aalten ⓘ https://orcid.org/0000-0002-1499-6908

### Decision letter and Author response

Decision letter https://doi.org/10.7554/eLife.90376.sa1
Author response https://doi.org/10.7554/eLife.90376.sa2

## Additional files

### Supplementary files

• MDAR checklist

### Data availability

Code used to generate Figure 2 - Supplement 1 A and C is deposited in GitHub (copy archived at *Czajewski, 2024*). Upon publication, information regarding novel genotypes and phenotypes will be deposited in FlyBase. Newly generated genotypes are available upon request.

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
