## [Editor Report]

This important study describes a model for O-GlcNac transferase (OGT) associated Intellectual Disability in *Drosophila*. The authors present convincing data showing that OGT mutant *Drosophila* exhibit defects in neuronal arborisation in development and behaviour (sleep) in adults. These results will be of interest to researchers and clinicians working on protein modifications and intellectual disability.

---

## [Decision Letter]

[Editors' note: this paper was reviewed by Review Commons.]

Thank you for submitting your article "Rescuable sleep and synaptogenesis phenotypes in a *Drosophila* model of O-GlcNAc transferase intellectual disability" for consideration by *eLife*. Your article has been reviewed by 3 peer reviewers at Review Commons, and the evaluation at *eLife* has been overseen by a Reviewing Editor and K VijayRaghavan as the Senior Editor. We are sorry about the delay.

Based on your manuscript, the reviews and your responses, we invite you to submit a revised version.

Regarding the revision plan from Review Commons, we notice in the rebuttal, that the authors note a few points that they do not plan to address. These are important to address however.

1) RL2 antibody not recognising low molecular weight proteins. This limitation should be described in the text of the manuscript.

2) Neuron vs muscle specific rescue of OGT. This is not a hard experiment. We would like the authors to proceed with it.

3) Inconsistencies with prev literature re: OGA knockdown bouton phenotypes. This also should be addressed in the text of the manuscript.

Other comments on the sleep data:

Major concern: sleep time phenotype reported for sxcC941Y

In Figure 5B they sleep less, in Figure 6A they don't. It is critical that this discrepancy be resolved.

[Editors' note: further revisions were suggested prior to acceptance, as described below.]

Thank you for submitting your article "Rescuable sleep and synaptogenesis phenotypes in a *Drosophila* model of O-GlcNAc transferase intellectual disability" for consideration by *eLife*. Your article has been reviewed by 3 peer reviewers at Review Commons, and the evaluation at *eLife* has been overseen by a Reviewing Editor and K VijayRaghavan as the Senior Editor.

Based on the previous reviews and the revisions, the manuscript has been improved but there are some remaining issues that need to be addressed, as outlined below:

In the new Figure 7, sxc is knocked down with RNAi in neurons (with elav GAL4) and glia (with repo GAL4). However, the experimental line (i.e. GAL4 > RNAi) is compared with only one heterozygous parental control the GAL4 / +. It is very important to compare the experimental line to both parental controls i.e. GAL4 / + and UAS sxc RNAi/ +. This is particularly important as the sleep phenotypes are modest. These experiments should not be hard to do.

[Editors' note: further revisions were suggested prior to acceptance, as described below.]

Thank you for submitting your article "Rescuable sleep and synaptogenesis phenotypes in a *Drosophila* model of O-GlcNAc transferase intellectual disability" for consideration by *eLife*.

Based on the previous reviews and the revisions, the manuscript has been improved but there are some remaining issues that need to be addressed, as outlined below:

We would like to see sleep of repo GAL4 / + , UAS sxc RNAi / +, and repo GAL4 > sxc RNAi in the same figure, done at the same time. This is standard presentation in the literature. In the latest revision, the authors present sleep of repo GAL4 / + and repo GAL4 > sxc RNAi in Figure 7, and in a separate experiment in Figure 7, supplement 1, they present UAS sxc RNAi / + and repo GAL4 > sxc RNAi. We would like to see the two heterozygous parental controls and the experimental group done at the same time. This is important as sleep time is variable, and even CantonS wild type flies, or isogenised flies for that matter, tested weeks apart will show variations in sleep time of the order of ~100 min / day. This experiment is simply looking at baseline sleep, so it should be fairly straightforward.

---

## [Author Response]

We would like to thank the reviewers for their careful reading of our manuscript and constructive comments.

Reviewer 1Indeed, the manuscript describe the alteration of total brain O-GlcNAc levels, but understanding pathways or protein specific changes would allow to identify the mechanisms potentially at the basis of the development of intellectual disability.

While finding the pathways involved in phenotypes described here is beyond the scope of the present manuscript, we have now performed RNAi experiments to dissect what cell types are responsible for the sleep phenotype observed in *sxc* mutant flies. These results have been incorporated in the revised manuscript (Results: line 415-480, Discussion: lines 625-629, Methods: lines 672-678, and a new Figure 7 detailing experiments involving DAM sleep assays performed on flies expressing *sxc* RNAi in glia or neurons).

Reviewer 32) Lacing fly food with compounds can sometimes lead to phenotypes not actually caused by the drug. There are reports I have previously seen where the compound can make the food more aversive or attractive, both leading to results not due to the drug. Specifically, it has been previously reported that starved flies (if the compound leads to aversion from the food and causes starvation) will reduce the bouts of sleep in *Drosophila* (Masek et al. J Exp Biol 2014; Figure 4). Do the authors know if the TMG treated food eaten at the same level as normal food? Is there the potential for a starvation phenotype?

We appreciate the need for this additional control experiment and have addressed this by measuring male adult ingestion of Thiamet G laced food by adding Blue No. 1 dye and measuring absorbance of lysed flies, as previously described in Wong et al. 2009 (PMID: 19557170). (Alterations to text: Results lines 409-411, Methods lines 732-741and additional panel D in figure S2).

*Description of the revisions that have already been incorporated in the transferred manuscript:*

Reviewer 1In figure 1C the blot show a different MW range compared to blots 1A and 1B, author should correct. “ and “For figure 1 and 2 the dot graph are too small and difficult to read”

The figures have been amended to address this.

Reviewer 3In the methods – Neuromuscular Junction Immunohistochemistry – which muscles and which types of boutons were imaged was not denoted in this section – it is described in results (lines 210-211) but should be in methods for ease to the reader.

The methods section has been amended.

The statistics and data analyses are some of the best I have seen to date. One concern is the removal of a single outlier data point described at line 575. Was this necessary? Does it change the data? If not, I would recommend leaving it in. If it does, I would further recommend additionally biasing toward the alternative hypothesis by additionally removing the data point that lies furthest from the outlier. This would reduce bias.

Removal of an outlier does indeed change the results of the data. Following the suggestions of the reviewer, we re-analyzed our data removing the minimum for the group for which we previously removed an outlier (the maximum).

1) line 391 mentions that feeding higher doses of TMG results in a non-rescue phenotype. Is there any data to support this statement (maybe supplementally) to give the reader the full picture of the availability of this compound? For example, how far above 250 μm does this happen?

This statement refers to adult Thiamet G feeding experiments, and the data to support this statement can be found in figures 2B and S2A. This statement has been amended for clarity and to include the caveat that even higher doses of TMG were not trialed.

*Description of analyses that authors prefer not to carry out:*

Reviewer 1Authors employed RL2 antibody for O-GlcNac detection, however it recognized mainly high MW proteins and it would be nice to obtain the alteration profile of low MW proteins at the same conditions.

We agree that the use of a single method for detecting O-GlcNAcylation is limited, however, there is no reason to believe that immunoblotting using this antibody would bias the interpretation of the effects of mutations studied here on global O-GlcNAcylation. Specifically, there is no reason to believe low molecular weight proteins are recognized and modified by OGT differently to high molecular weight proteins. While gaining insight into substrate specific alterations in O-GlcNAcylation is of great interest to us, this is technically very challenging and beyond the scope of this study.

Reviewer 2… would it be possible for the authors to overexpress specifically in neurons wildtype OGT postnatally on a mutant background and quantify the effects on neuro-muscular synapse number and morphology? It would be interesting to compare these data with a similar experiment where they overexpress wildtype OGT in the corresponding muscle.

While temporal control of transgene expression is possible in *Drosophila,* it is not a technique that we routinely use and would require extensive optimization to include in the present manuscript.

Reviewer 3In Figure 3D the authors show sxcWT compared with OgaKO with no significant difference at ~20 boutons in the count. Other work done by [47] in their reference list (ref 47: Figure 2D) shows an increase in OgaKO boutons vs WT and also shown in [50] (ref 50; Figure 4B) where # of boutons in 1B muscle 4 is increased in OgaKO significantly. There appears to be a difference in what was found with OgaKO vs controls in the authors' results vs these two manuscripts and it should be noted and explained to the reader.

This is indeed an inconsistency we have observed, however, looking at reference Fenckova et al. 2022 (47 in our manuscript) we find that in figure legend 2 the following is stated: “None of the parameters is significantly affected in the *Oga^KO^* larvae (N = 30, in purple; *Oga^KO^* experiments were performed simultaneously and first published here [53] with significantly increased bouton counts (p <0.05) without multiple testing correction)” Reference [53] in the quote refers to Muha et al. 2020 (reference 50 in our manuscript). Therefore, it appears that this effect is too weak to withstand multiple correction testing, which we employ in our analysis.

[Editors’ note: what follows is the authors’ response to the second round of review.]

Regarding the revision plan from Review Commons, we notice in the rebuttal, that the authors note a few points that they do not plan to address. These are important to address however.1) RL2 antibody not recognising low molecular weight proteins. This limitation should be described in the text of the manuscript.

While it is known that there are biases to detection of O-GlcNAcylation by RL2, there is no evidence to suggest that this is based on the molecular weight of the protein. The limitations of using RL2 to quantify O-GlcNAcylation levels have been addressed in text (Lines 494-499).

2) Neuron vs muscle specific rescue of OGT. This is not a hard experiment. We would like the authors to proceed with it.

Experiments to address this point have been performed and have now been incorporated into the text of the manuscript (Results: lines 286-292, Discussion: 589-593 and 605-608, Methods: lines 667-672, Supplementary Figure 4).

3) Inconsistencies with prev literature re: OGA knockdown bouton phenotypes. This also should be addressed in the text of the manuscript.

With regard to inconsistencies with previous literature on *Oga* knockout bouton phenotypes, additional text has been added to highlight that reports of *Oga* knockout phenotypes at the NMJ are themselves inconsistent (Discussion: lines 585- 589).

4) Major concern: sleep time phenotype reported for sxcC941Y In Figure 5B they sleep less, in Figure 6A they don't. It is critical that this discrepancy be resolved.

This discrepancy has been addressed in lines 392-394 (Results section) and 618-625 (discussion) and has been attributed to differences in experimental design between results shown in Figure 5 and Figure 6. The most salient difference in experimental design is in the food used in these sleep assays, which was altered for the pharmacological rescue experiments to accommodate the addition of TMG. Standard *Drosophila* food requires boiling to dissolve in water, which hinders the addition of temperature sensitive compounds.

[Editors’ note: what follows is the authors’ response to the third round of review.]

Based on the previous reviews and the revisions, the manuscript has been improved but there are some remaining issues that need to be addressed, as outlined below:In the new Figure 7, sxc is knocked down with RNAi in neurons (with elav GAL4) and glia (with repo GAL4). However, the experimental line (i.e. GAL4 > RNAi) is compared with only one heterozygous parental control the GAL4 / +. It is very important to compare the experimental line to both parental controls i.e. GAL4 / + and UAS sxc RNAi/ +. This is particularly important as the sleep phenotypes are modest. These experiments should not be hard to do.

To control for the effect of genetic background or leaky RNAi transgene expression, an additional experiment was performed to compare total sleep amount and architecture between *repo>sxc* RNAi and +/UAS:*sxc* RNAi flies. Analysis was restricted to this comparison as no effect was seen when the RNAi transgene was expressed in neurons (*elav>* sxc RNAi). The results of this experiment were appended to Figure 7 as a figure supplement (Figure 7 Supplement 1) and are referenced in lines 361-363 of the Results section.

[Editors’ note: what follows is the authors’ response to the fourth round of review.]

Based on the previous reviews and the revisions, the manuscript has been improved but there are some remaining issues that need to be addressed, as outlined below:We would like to see sleep of repo GAL4 / + , UAS sxc RNAi / +, and repo GAL4 > sxc RNAi in the same figure, done at the same time. This is standard presentation in the literature. In the latest revision, the authors present sleep of repo GAL4 / + and repo GAL4 > sxc RNAi in Figure 7, and in a separate experiment in Figure 7, supplement 1, they present UAS sxc RNAi / + and repo GAL4 > sxc RNAi. We would like to see the two heterozygous parental controls and the experimental group done at the same time. This is important as sleep time is variable, and even CantonS wild type flies, or isogenised flies for that matter, tested weeks apart will show variations in sleep time of the order of ~100 min / day. This experiment is simply looking at baseline sleep, so it should be fairly straightforward

We thank the reviewers for their comment, we agree that the data were not presented in a manner that is standard in the field. This has been amended so that all the genotypes analysed for the final Results section of the manuscript have been analysed together (thereby including appropriate multiple testing correction) and Figure 7 has been altered to represent all the genotypes together.